# Deep learning-based polygenic risk analysis for Alzheimer's disease prediction

Xiaopu Zhou [1,2,3], Yu Chen[1,3,4], Fanny C. F. Ip[1,2,3], Yuanbing Jiang [1,2], Han Cao[1], Ge Lv[5], Huan Zhong[1,2], Jiahang Chen[5], Tao Ye [1,3,4], Yuewen Chen[1,3,4], Yulin Zhang[3], Shuangshuang Ma[3], Ronnie M. N. Lo[1], Estella P. S. Tong[1], Alzheimer's Disease Neuroimaging Initiative*, Vincent C. T. Mok [6], Timothy C. Y. Kwok [7], Qihao Guo[8], Kin Y. Mok[1,2,9,10], Maryam Shoai[9,10], John Hardy[2,9,10,11], Lei Chen[5], Amy K. Y. Fu[1,2,3] & Nancy Y. Ip [1,2,3✉]

**Abstract**

**Background** The polygenic nature of Alzheimer's disease (AD) suggests that multiple variants jointly contribute to disease susceptibility. As an individual's genetic variants are constant throughout life, evaluating the combined effects of multiple disease-associated genetic risks enables reliable AD risk prediction. Because of the complexity of genomic data, current statistical analyses cannot comprehensively capture the polygenic risk of AD, resulting in unsatisfactory disease risk prediction. However, deep learning methods, which capture nonlinearity within high-dimensional genomic data, may enable more accurate disease risk prediction and improve our understanding of AD etiology. Accordingly, we developed deep learning neural network models for modeling AD polygenic risk.

**Methods** We constructed neural network models to model AD polygenic risk and compared them with the widely used weighted polygenic risk score and lasso models. We conducted robust linear regression analysis to investigate the relationship between the AD polygenic risk derived from deep learning methods and AD endophenotypes (i.e., plasma biomarkers and individual cognitive performance). We stratified individuals by applying unsupervised clustering to the outputs from the hidden layers of the neural network model.

**Results** The deep learning models outperform other statistical models for modeling AD risk. Moreover, the polygenic risk derived from the deep learning models enables the identification of disease-associated biological pathways and the stratification of individuals according to distinct pathological mechanisms.

**Conclusion** Our results suggest that deep learning methods are effective for modeling the genetic risks of AD and other diseases, classifying disease risks, and uncovering disease mechanisms.

**Plain language summary**

Polygenic diseases, such as Alzheimer's disease (AD), are those caused by the interplay between multiple genetic risk factors. Statistical models can be used to predict disease risk based on a person's genetic profile. However, there are limitations to existing methods, while emerging methods such as deep learning may improve risk prediction. Deep learning involves computer-based software learning from patterns in data to perform a certain task, e.g. predict disease risk. Here, we test whether deep learning models can help to predict AD risk. Our models not only outperformed existing methods in modeling AD risk, they also allow us to estimate an individual's risk of AD and determine the biological processes that may be involved in AD. With further testing and optimization, deep learning may be a useful tool to help accurately predict risk of AD and other diseases.

A full list of author affiliations appears at the end of the paper.

Recent biotechnological advances have expanded the study of human diseases from descriptive to quantitative analyses. In particular, genomic variations are a major category of risk factors that contribute to various human diseases[1]. As genetic variants are present in the human genome, they are a reliable source of information for disease risk prediction throughout life[2]. Therefore, profiling genetic variation enables disease risk prediction in individuals before disease onset, which is especially valuable for clinical investigations and developing intervention strategies for age-related diseases such as Alzheimer's disease (AD)[3–5].

AD is one of the most common neurodegenerative diseases and is highly prevalent in older populations (~10% among people ≥65 years old)[6]. Genetic factors play a pivotal role in AD pathogenesis, supporting the utility of genetic information in AD risk prediction[7,8]. In addition, developing effective genetic screening tools for the early prediction of AD is vital for disease management[9]. However, recent genome-wide association studies (GWASs) have revealed that AD is polygenic in nature, with dozens of loci contributing to disease risk[10–19]. APOE-ε4 is the most prevalent genetic risk factor for AD[20]. However, as other common AD-associated variants exert small to moderate effects on AD risk, they cannot be used individually to infer disease risk[10–19]. Therefore, to determine an individual's risk of developing AD, we need to develop models that encompass multiple informative genetic variants.

Tremendous efforts have been made to develop polygenic score models using genetic information to estimate disease risk[21,22]. One of the most commonly used is the weighted polygenic risk score (PRS) model, which predicts an individual's risk of disease by summarizing the risk effects of multiple variants obtained from GWASs[23]. Numerous studies have investigated the utility of the weighted PRS model for classifying patients with various diseases[20,24–27]. In particular, the weighted PRS model can be used to classify clinically and pathologically confirmed AD as well as predict the onset age of AD. Collectively, these findings highlight the applicability of polygenic score models for predicting disease risk, particularly for AD[28–30].

A weighted PRS model is constructed by multiplying the weighted sum of risk allele dosages by their corresponding effect sizes, which are derived from GWASs. However, most GWASs calculate the effect sizes of each variant independently without considering epistatic effects (i.e., the effects of interaction among the variants), resulting in an inaccurate estimation of an individual variant's contribution to the disease[31–34]. Although various modified weighted PRS models have been proposed[35–37], they have not been thoroughly tested using real-world data and are unlikely to adapt well to high-dimensional genomic data owing to their low model complexity (i.e., insufficient number of model parameters).

Notably, recent studies suggest the possible application of statistical learning (e.g., lasso [least absolute shrinkage and selection operator])[38] and deep learning (e.g., neural network) models[39,40] for polygenic risk analysis and disease risk classification. Specifically, as neural network models have higher model complexity (i.e., a greater number of model parameters) as well as sophisticated and multilayered architecture, they may be better suited to handle high-dimensional genomic data for disease classification than weighted PRS models. Nevertheless, the performance of lasso and neural network models for AD polygenic risk prediction has not been systematically evaluated. Therefore, it is of interest to investigate whether deep learning models, particularly neural network models, can be used for polygenic risk analysis and AD risk classification.

In this study, we aimed to develop neural network models for modeling AD polygenic risk. In particular, we find that neural network models are effective for classifying patients with AD, outperforming both weighted PRS and lasso models. Furthermore, by combining the predicted risk scores derived from neural network models with AD-associated endophenotypic data, we identify potential pathological mechanisms that contribute to AD polygenic risk. Together, our results suggest that deep learning methods can be used to predict AD risk, stratify at-risk individuals into subgroups, and identify the mechanisms underlying the disease.

## Methods

**Study data.** To investigate the performance of polygenic score models for classifying AD risk, we included the array data from the National Institute on Aging Alzheimer's Disease Centers (ADC) cohort (phs000372.v1.p1), the Late Onset Alzheimer's Disease Family Study cohort ("LOAD cohort" hereafter, phs000168.v2.p2), and the Alzheimer's Disease Neuroimaging Initiative cohort (ADNI) cohort (http://adni.loni.usc.edu/) in our analysis. The demographic data of these cohorts are presented in Supplementary Table 1. The details of the quality control and imputation processes are presented in the Supplementary Methods.

We included two Chinese whole-genome sequencing (WGS) cohorts to study the polygenic score models. The data for Chinese WGS cohort 1 ($N = 2340$ comprising 1116 patients with AD, 309 patients with mild cognitive impairment [MCI], and 915 age- and sex-matched normal controls [NCs]) have been published[18]. The data for Chinese WGS cohort 2 ($N = 1077$ comprising 356 patients with AD, 68 patients with MCI, and 653 age- and sex-matched NCs) have also been published[38]. The phenotypic data of the participants analyzed in this study were based on the participants' most recent diagnostic records (as of December 2019). The study was approved by the Clinical Research & Ethics Committees of Joint Chinese University of Hong Kong-New Territories East cluster for Prince of Wales Hospital (CREC Ref no. 2015.461), Kowloon Central Cluster/Kowloon East Cluster for Queen Elizabeth Hospital (KC/KE-15-0024/FR-3), and Human Participants Research Panel of the Hong Kong University of Science and Technology (CRP#180 and CRP#225). All participants provided written informed consent for both study participation and sample collection.

**Variant selection for model construction.** We selected variants to evaluate the polygenic score models based on the AD GWAS summary statistics reported by Jansen et al.[13]. For model construction, we applied three different $p$-value thresholds (<1E−8, <1E−6, and <1E−4) to the resultant variants. We retained the variants detected by all imputed array data from the ADC, LOAD, and ADNI cohorts that also fell into the corresponding $p$-value ranges for model construction and comparison (selected according to single nucleotide polymorphism [SNP] ID).

For polygenic score analysis in the European-descent datasets, we compared the performance of the weighted PRS, lasso, and neural network models in three different scenarios: (i) using all the data from the three AD cohorts (i.e., ADC, LOAD, and ADNI) as training data; (ii) using all the data from the three AD cohorts for five-fold cross-validation analysis; and (iii) using two AD cohorts (i.e., ADC and LOAD) as training data and the remaining cohort (i.e., ADNI) as validation data.

For (ii), we conducted the five-fold cross-validation 10 times. We preclassified the samples using the *createFolds* function from the *caret* package in R; the classified labels were stored in a text file to allow for a fair comparison with other models (i.e., the weighted PRS and neural network models) for classification accuracy. For (iii), we used the data of 70% of individuals from

the LOAD ($n = 2995$ of 4278) and ADC ($n = 3984$ of 5692) cohorts for model training and used the data of the remaining 30% ($n = 2991$ of 9970 total) to evaluate model accuracy at the end of each epoch. We used the data from the ADNI cohort ($N = 1382$ comprising 689 patients with AD and 693 NCs) as a cross-evaluation dataset to assess the performance of polygenic score models. Of note, to further avoid overestimating the model performance in the validation dataset rendered by potential duplicate samples among the three AD cohorts, we conducted identity-by-descent analysis using PLINK. We found 415 potential duplicate samples (PI_HAT > 0.90; $n = 16$, 348, and 51 for ADC, LOAD [identified and removed from the data], and ADNI, respectively) and then reconstructed the model and tested its performance.

**Construction and testing of the weighted polygenic risk score model**. We constructed the weighted PRS model by elementwise multiplication of allele dosage and selected the corresponding effect sizes from the AD GWAS summary statistics[13] for the variants according to the p-value thresholds. We applied the same calculation method to evaluate the reported model for the Chinese population[11]. We calculated weighted PRSs in *RStudio* (v1.3.1056) using R programming (v4.0.2). We further used the effect sizes of the selected variants from another set of AD GWAS summary statistics (i.e., the IGAP 2019 Rare Variant Analysis stage 1 dataset)[12] to generate a parallel weighted PRS model. More than 96% of the variants selected in the first GWAS were captured by the second GWAS for model construction.

In addition, we used LDpred[35], Winner's curse correction[36], AnnoPred[37], and SBayesR[35] to model polygenic risk according to the instructions in each program's user manual. We ran LDpred, AnnoPred, and SBayesR on the variant lists before the linkage disequilibrium (LD)-clumping steps, as these applications can utilize LD information for PRS modeling. Based on a p-value threshold of 1E−4, 1149 sites were excluded by LDpred because of its built-in filtering criteria (which removes all A/T and G/C SNPs), and 6860 sites were excluded by AnnoPred because the software was designed to only take variants listed in the HAPMAP3 dataset. AnnoPred produced no output among the four programs, possibly because too many variants were filtered. For model evaluation in the scenarios (i) (i.e., all samples) and (ii) (i.e., five-fold cross-validation) scenarios, we recalculated the effect sizes by conducting logistic regression—adjusting for the confounding effects of age, sex, and genomic structure (represented by the top five principal components)—in each training set. As for scenario (iii) and all remaining weighted PRS analyses, we obtained the effect sizes from the AD GWAS results reported by Jansen et al. to provide a more unbiased estimation on variant effects because of the large sample size.

For the Chinese data, we obtained the effect sizes from meta-analysis results of two Chinese datasets. We also used another published AD GWAS in the Chinese population to evaluate model performance for classifying AD risk[18].

**Construction and testing of the lasso model**. We applied logistic regression to regress out the potentially confounding effects of age, sex, and genomic structure (represented by the top five principal components). We constructed the logistic lasso regression model using the *cv.glmnet()* function from the *glmnet* package, with five-fold cross-validation (*alpha* = 1, *type.-measure* = "mse", *nfolds* = 5) for the variants selected according to the p-value thresholds[38]. We selected the λ-value that retained the most variants for the risk score calculation and used the *predict()* function to retrieve the polygenic scores.

Regarding the Chinese WGS datasets, we used Chinese WGS cohort 1 for the training dataset; we applied the same approach for model construction using the information from the 37 variants selected by the association test (regressing out the potentially confounding effects of age, sex, and genomic structure). We subsequently used Chinese WGS cohort 2 to evaluate the resultant model. We performed all analyses of the lasso polygenic score model in *RStudio* (v1.3.1056) using R programming (v4.0.2). We fixed the value of the random seed to the same constant value before performing all analyses.

**Construction and testing of the neural network model**. We constructed the neural network model using the *Sequential()* function from the *Keras* package, an API for TensorFlow. Before performing analyzing the European-descent population, we annotated the selected SNPs for their associated loci using *ANNOVAR*[41] (77, 141, and 696 loci for p-value thresholds <1E−8, <1E−6, and <1E−4, respectively). We designed a seven-layer model, with the first and third layers as dropout layers (*dropout rate* = 0.2 or 0.3). We designed the number of nodes as follows: 3 × number of loci (based on the assumption that a maximum of three different haplotypes are associated with AD in each locus), 1 × number of loci (corresponding to the locus number), 22 (corresponding to the chromosome number), 5 (an arbitrary number corresponding to the potential number of pathways that affect the disease in parallel), and 1 (corresponding to the risk score). We applied exponential decay using the *ExponentialDecay()* function to accelerate the analysis (*decay steps* = 100,000, *decay rate* = 0.96, *staircase* = True) with the sigmoid function as the activation function. We applied the binary *crossentropy loss* function and evaluated model accuracy using the *accuracy*, *auROC*, and *auPRC* metrics. We applied the neural network models for polygenic score analysis in the European-descent population for three scenarios as described in the previous section: (i) no validation, (ii) five-fold cross-validation, and (iii) validation using an independent cohort. For (i), we configured the models with 2000 epochs, a batch size of 256, and a learning rate of 0.5. For (ii), we configured the models with 1500 epochs, a batch size of 1024, and a learning rate of 0.5. For (iii), we chose the number of epochs (i.e., 500–800) by observing the model performance plot for the training and validation datasets. We further applied the early stopping using the *EarlyStopping()* function (*patience* = 50 or 100 epochs) when examining the transethnic performance of the neural network model (i.e., training models on Chinese data before applying them to European-descent data, or vice versa).

During model training for the Chinese WGS datasets, we used Chinese WGS cohorts 1 and 2 as the training and validation datasets, respectively. Accordingly, we designed a seven-layer model for the study, with the second and fourth layers as dropout layers (*dropout rate* = 0.3); the numbers of nodes in each layer were 50, 30, 10, 5, and 1. We applied the binary *crossentropy loss* function and evaluated model accuracy using the *accuracy*, *auROC*, and *auPRC* metrics. We configured the models with a batch size of 256. We chose the number of epochs (i.e., 1000) by observing the model performance plot for the training and validation datasets. Moreover, we used the *backend.function()* function to extract the outputs from the nodes from the penultimate layer for further analyses. We fixed the value of the random seed to the same constant value before conducting all analyses.

**Construction and testing of the graph neural network model**. We modeled disease risk as a graph classification problem. In brief, each participant was represented in a graph with nodes

denoting the 37 selected variants, edges denoting pairwise LD (calculated by PLINK) among the variants, and graph labels denoting phenotypes. For node features, in addition to the allele dosage, we considered the biological properties of variants, including whether they resided in coding or untranslated regions, and the number of events of histone, open chromatin, polymerase, and transcription factor binding. We retrieved these biological properties of variants from the SNPnexus database[42]. Considering the possible variations of LD among ethnic backgrounds, we first inferred the ethnic backgrounds of 11,352 participants from the ADNI, ADC, and LOAD cohorts using GRAF-pop software[43]. Accordingly, for each ethnic group, we obtained the LD for 37 variants using the 1000 Genomes Project Phase 3 data[44].

To construct the graph neural network model, all node features were normalized dimension-wise. We used an $R^2$ threshold of 0.6 to determine if two variants connected and created the adjacent matrix for the edge information. In particular, for each individual, we used the LD data obtained from the matched ethnic background to construct the adjacent matrix. We followed the common practice[45] of training a three-layer graph convolutional network[46] with 128 hidden dimensions. We used two fully connected layers as the final classifier with 64 dimensions. We adopted Relu as the nonlinear function and employed *global max pooling*. We implemented the model using *Pytorch*[47] and trained it using *Adam* optimizer[48] with an early stop (*patience step* = 20).

**Whole-genome sequencing**. We performed WGS (5× coverage) using Novogene. We sequenced the genomic DNA libraries on an Illumina HiSeq × Ten and NovaSeq platform (San Diego, CA, USA) (150-bp paired-end reads). We adopted the GotCloud pipeline[49] to detect variants from our low-pass WGS data. In brief, we subjected sequencing data to FastQC (https://www.bioinformatics.babraham.ac.uk/projects/fastqc/) for quality control and Trimmomatic[50] to trim and filter low-quality reads. We mapped clean data to the GRCh37 reference genome containing decoy fragments using BWA-mem. We conducted subsequent analysis by subjecting data to the GotCloud pipeline with data processing and variant detection using the default settings[18]. We then subjected the clean genotype files to Beagle[51] for genotyping refinement. For Chinese WGS cohort 2, we also used Thunder[52] for genotyping refinement after Beagle processing.

**Analysis of plasma protein and brain imaging data**. Next, we analyzed plasma protein and brain imaging data collected from a subgroup of participants from Chinese WGS cohort 2[38]. Specifically, we analyzed plasma amyloid-beta (Aβ)$_{42}$, Aβ$_{40}$, tau, and neurofilament light polypeptide (NfL) levels in 157 patients with AD and 125 NCs by single-molecule detection assay (Neurology 3-Plex A Advantage Kit, #101995; NF-light Advantage Kit, #103186; Quanterix, Billerica, MA, USA). We also examined plasma p-tau181 levels in 154 patients with AD and 118 NCs (pTau-181 Advantage V2 Kit, #103714, Quanterix, Billerica, MA, USA). Detection was performed at the Quanterix Accelerator Lab (Boston, MA, USA). Moreover, the plasma samples of 97 patients with AD and 69 NCs from Chinese WGS cohort 2 were further subjected to Olink Proteomics (Boston, MA, USA) to determine the abundance of 1,160 plasma proteins by proximity extension assay. The following panels were used for the analysis: Cardiometabolic (91802), Cardiovascular II (91202), Cardiovascular III (91203), Cell Regulation (91702), Development (91703), Immune Response (91701), Inflammation (91301), Metabolism (91801), Neuro Exploratory (91502), Neurology (91501), Oncology II (91402), Oncology III (91403), and Organ Damage (91901).

For brain imaging analysis, we retrieved T1-weighted magnetization-prepared rapid acquisition with gradient-echo (MPRAGE) and fluid-attenuated inversion recovery (FLAIR) sequences for 78 patients with AD and 104 NCs from Prince of Wales Hospital (Hong Kong, China). We deidentified the raw imaging files and sent them to BrainNow Medical Technology (Hong Kong, China) to analyze volumetric information in different brain regions and white matter hyperintensity levels. We did not perform multiple test adjustment because of the limited sample sizes of the plasma protein and brain magnetic resonance imaging data.

**Statistical analysis**. We performed a meta-analysis with a fixed-effects model using *METASOFT* software (v2.0.0) for variant analysis. We evaluated the classification accuracy of the models by calculating the area under the receiver operating characteristic curve (auROC) using the *roc()* function from the *pROC* package or the area under the precision–recall curve (auPRC) using the *pr.curve()* function from the *PRROC* package. We estimated the 95% confidence intervals of the auROC using bootstrap methods from the *ci.auc()* function. We used the *roc.test()* function using the bootstrap method to test the potential differences in auROCs obtained from the different models. We determined the low-, medium-, and high-risk groups by fitting risk scores to a Gaussian mixture model using the *normalmixEM()* function from the *mixtools* package for the patients with MCI in Chinese WGS cohort 1 ($k = 3$, maxit = 200, ECM = T). We calculated the probability of an individual being classified into the low- or high-risk group by using the corresponding scores as the input for the fitted probability distributions (using values of $\mu$ and $\sigma$ from the fitted Gaussian mixture model). Meanwhile, we calculated the probability of an individual being classified into the medium-risk group by calculating the difference between 1 and the sum of the probabilities of being in the low- or high-risk group.

In addition, we performed an association analysis between polygenic score or risk group and disease phenotype by logistic regression using the *glm()* function from the *stats* package. We also performed an association analysis between polygenic score or risk/phenotype group and cognitive performance, plasma biomarkers, and brain volume using a robust linear regression model using the *lmrob()* function from the *robustbase* package, with age, sex, and genomic structure (represented by the top five principal components) as covariates. For cognitive performance, we applied rank-based, inverse-normal transformation to the cognitive scores using the *RankNorm()* function from the *RNOmni* package, with age, sex, and genomic structure (represented by the top five principal components) as covariates. Regarding the brain imaging data, we further included intracranial volume as a covariate to normalize the possible interindividual variation in brain volume. We conducted the Spearman's rank correlation test using the *cor.test()* function in R to examine the performance of models constructed from different ethnic backgrounds.

We performed a Gene Ontology enrichment analysis of the UniProt IDs in the Database for Annotation, Visualization and Integrated Discovery (DAVID) (https://david.ncifcrf.gov/). Moreover, we performed a protein–protein interaction network analysis of the UniProt IDs in the Search Tool for the Retrieval of Interacting Genes/Proteins (STRING) database (https://string-db.org/). For our cluster analysis, we applied *k*-means clustering to separate plasma proteins into individual clusters using the *kmeans()* function from the *stats* package for the absolute values of *t*-statistics obtained from association tests between the levels of plasma proteins (i.e., normalized protein expression) and the polygenic scores from the five different modules. We determined the optimal number of clusters by using the elbow method

implemented in the *fviz_nbclust()* function from the *factoextra* package.

For the cell-type enrichment analysis, we obtained the gene expression levels measured by RNA sequencing in individual blood cell types from the BLUEPRINT database (http://dcc.blueprint-epigenome.eu/). We performed a cell-type enrichment analysis using the *TissueEnrich* package. Briefly, we subjected cell-type-specific transcript levels measured as fragments per kilobase per million mapped fragments (FPKM) for 1159 plasma proteins to the *teGeneRetrieval()* function to first identify genes expressed by specific groups of cells (*foldChangeThreshold = 1.5, expressedGeneThreshold = 5*). We then used the *teEnrichmentCustom()* function for the enrichment analysis to identify the specific cell types associated with individual gene clusters (*tissueSpecificGeneType = 1*).

To stratify participants according to the outputs from the penultimate layer in the neural network model, we determined the optimal number of groups by using the elbow method implemented in the *fviz_nbclust()* function from the *factoextra* package. Then, we used the *umap()* function from the *umap* package in R to apply the uniform manifold approximation and projection (UMAP) method to project individual participants onto the two-dimensional plane for visualization. To examine individual variants' contributions to the polygenic score, we conducted a partial correlation analysis using the *pcor()* function (method = "spearman") from the *ppcor* package in R. We performed the annotation of variant functions by submitting the SNP rsID to the SNPnexus database (https://www.snp-nexus.org/v4/citation/)[42].

**Data visualization**. We generated a schematic diagram of the study using Microsoft PowerPoint (v2105). We generated heatmaps of AD classification accuracy (i.e., auROC and auPRC), box plots, volcano plots, bar charts, and dot plots using GraphPad Prism (v8.3.0). We plotted ROC and PR curves using the plot() function in R. Moreover, we generated histograms of polygenic scores using the *ggplot()* function from the *ggplot2* package with the *geom_density_ridges_gradient()* function from the *ggridges* package. We generated a heatmap to visualize distinct protein clusters using the *heatmap.2()* function from the *gplots* package. In addition, we visualized the protein–protein network using Cytoscape (v3.8.2) based on the node and interaction score information obtained from the STRING database (v11.0). We annotated candidate *cis*-regulatory regions and other epigenetic signatures using the SCREEN database (https://screen.encodeproject.org/)[53], and visualized transcription factor binding events using the University of California Santa Cruz Genome Browser (https://genome.ucsc.edu/)[54,55]. We also annotated the chromatin accessible regions from human brain single-cell ATAC-seq (Assay for Transposase-Accessible Chromatin using sequencing) data[56].

**Reporting summary**. Further information on research design is available in the Nature Portfolio Reporting Summary linked to this article.

## Results

To systematically evaluate the performance of different polygenic score models for AD risk classification, we obtain the genotype and phenotype data of three AD cohorts: the ADNI cohort[57], the LOAD cohort[58], and the ADC cohort[59,60] (N = 11,352 comprising 6681 patients with AD and 4671 NCs; Supplementary Table 1). For model construction, we select AD-associated variants from the AD GWAS summary statistics reported by Jansen et al[13]. with three *p*-value thresholds—1E−4, 1E−6, and 1E−8—

which yielded 8100, 2959, and 1799 SNPs, respectively. (Supplementary Table 2). Figure 1 shows a schematic flow diagram of the study.

**Evaluation of the weighted polygenic risk models for Alzheimer's disease risk prediction**. To examine the performance of the weighted PRS models for classifying AD risk, we calculate PRSs based on the effect sizes (i.e., weights) from the AD GWAS summary statistics reported by Jansen et al[13]. Besides including all the variants from that study, we also include the variants that reside outside the *APOE* locus (chr19:44000000-46000000; GRCh38)—the region that harbors the most prevalent risk factor for AD—to estimate their polygenic risk effects. Meanwhile, we apply LD-based clumping to obtain the minimum number of variants needed for classifying disease risk. We evaluate model performance by calculating the auROC and auPRC, with higher values indicating more accurate AD classification. We show that the weighted PRS model constructed from the variant set with the greatest number of variants (n = 8100; p < 1E−4) after LD-based clumping yields the highest classification accuracy (auROC: ~0.67; Supplementary Fig. 1; Supplementary Table 3). Moreover, only including the genetic variants outside of the *APOE* locus provide enough information for AD risk classification as suggested by auROCs from ~0.57 to 0.59 (Supplementary Fig. 1; Supplementary Table 3).

To assess the performance of different weighted PRS models for classifying AD risk, we conduct a parallel weighted PRS analysis (designated wPRS2) using the effect sizes from the summary statistics of the International Genomics of Alzheimer's Project (IGAP) 2019 Rare Variant Analysis stage 1 data[12]. We find no significant differences in the accuracy of AD risk classification scores generated from the two sets of the AD GWAS summary statistics (Supplementary Fig. 1; Supplementary Table 3). Meanwhile, we also construct modified weighted PRS models using different tools including LDpred[35], Winner's curse correction[36], AnnoPred[37], and SBayesR[35]. These modified models do not significantly improve the accuracy of AD risk classification (auROC: ~0.67; Supplementary Fig. 1; Supplementary Table 4). Together, these findings demonstrate that genetic information can be used for AD risk classification (auROC: ~0.67 from the weighted PRS models), providing a basis for further evaluation of the performance of neural network models for classifying AD risk.

**The neural network model outperforms both lasso and weighted polygenic risk score models for Alzheimer's disease risk prediction**. To evaluate the performance of a neural network model for predicting AD risk, we construct a seven-layer neural network model for disease risk classification with the same sets of variants used to construct the weighted PRS models (see above: 8100, 2959, and 1799 variants). In addition, we construct a lasso model to model polygenic risk in each scenario as a comparison, because we previously showed that polygenic scores derived from lasso models can be used for disease risk classification[38].

First, to examine the potential of using the three models for disease risk classification, we construct the models based on all data from the three AD cohorts (N = 11,352; Supplementary Table 1). We find that for all three models, including more SNPs in the model construction increased the accuracy of AD risk classification (Supplementary Fig. 2; Supplementary Table 5). In particular, when we include 8100 SNPs in the model construction, the prediction accuracy of the neural network model is nearly perfect (auROC = 1.00) and significantly higher than that of both the lasso (auROC = 0.94; p < 0.001) and weighted PRS models (auROC = 0.71; p < 0.001; Supplementary Fig. 2;

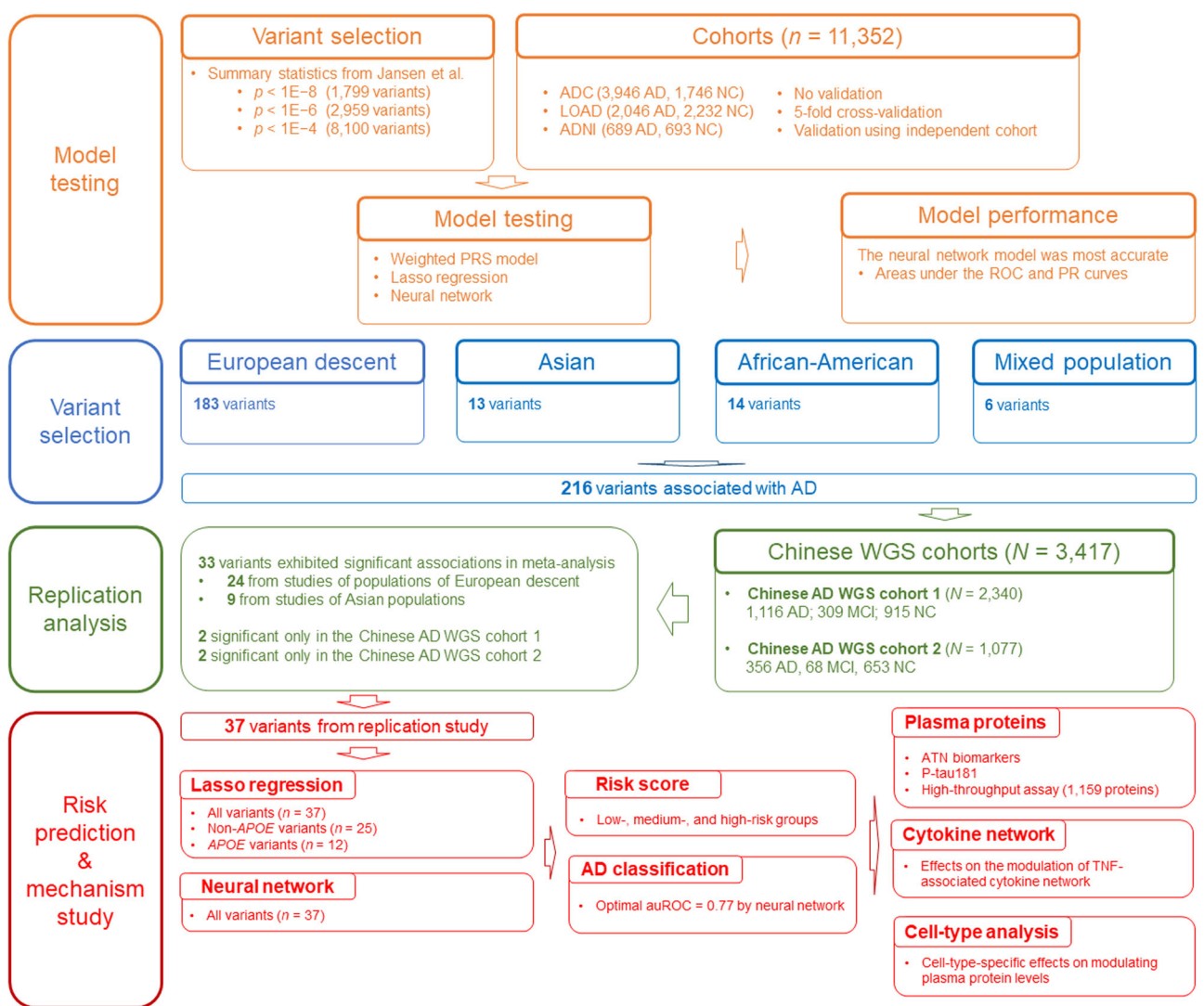

**Fig. 1 Study schematic.** Schematic diagram showing the study design. AD, Alzheimer's disease; ADC, National Institute on Aging Alzheimer's Disease Center cohort; ADNI, Alzheimer's Disease Neuroimaging Initiative cohort; ATN, amyloid-beta, tau, and neurofilament light polypeptide; auROC, area under the receiver operating characteristic curve; lasso, least absolute shrinkage and selection operator; LOAD, Late Onset Alzheimer's Disease Family Study cohort; MCI, mild cognitive impairment; $n$, number of samples or variants; NC, normal control; NN, neural network; PR, precision–recall; PRS, polygenic risk score; $p$, p-values; p-tau181, tau phosphorylated at threonine-181; ROC, receiver operating characteristic; TNF, tumor necrosis factor; WGS, whole-genome sequencing.

Supplementary Table 5). However, the high auROC values (>0.90) obtained from the neural network and lasso models suggest possible overfitting during the model training steps. Therefore, the model performance should be further evaluated and compared with samples that are independent of those used in the model training.

To mitigate overfitting, we conduct a five-fold cross-validation analysis that trains the model using 80% of the data and tests model performance with the remaining 20% of the data. Again, for all three models, including more SNPs improves AD risk classification accuracy. Moreover, when we use 8100 SNPs in the model construction, the neural network model (auROC = 0.73) exhibits greater prediction accuracy than both the lasso (auROC = 0.72; $p < 0.001$) and weighted PRS models (auROC = 0.69; $p < 0.001$; Supplementary Fig. 3; Supplementary Table 6). Therefore, our findings suggest that the neural network model predicts AD risk better than both the weighted PRS and lasso models.

To evaluate the performance of the three models (i.e., the weighted PRS, lasso, and neural network models) for predicting

disease risk across different cohorts, we train the models using 70% of the ADC and LOAD data and then evaluate and fine-tune the models using the remaining 30% of the data. We then validate the model performance in the ADNI dataset. Again, including more SNPs in the model construction achieves higher classification accuracy (Fig. 2a, b; Table 1; Supplementary Fig. 4; Supplementary Table 7). Moreover, when we use the same number of SNPs for the model construction, the neural network model outperforms the weighted PRS and lasso models for AD risk classification. For example, when we include 8,100 SNPs in the model construction, the auROCs for the weighted PRS, lasso, and neural network models in the ADC cohort is 0.70, 0.81, and 0.84, respectively; the auPRCs are 0.84, 0.89, and 0.92, respectively (Fig. 2a–d; Table 1; Supplementary Table 7). Of note, the neural network model also performs better for classifying AD risk than the other two models in the ADNI cohort (as suggested by higher auPRC values) (Fig. 2; Table 1; Supplementary Table 7). Moreover, to avoid over-estimating the model performance, we remove potential duplicate samples ($n = 415$) inferred by our identity-by-descent analysis

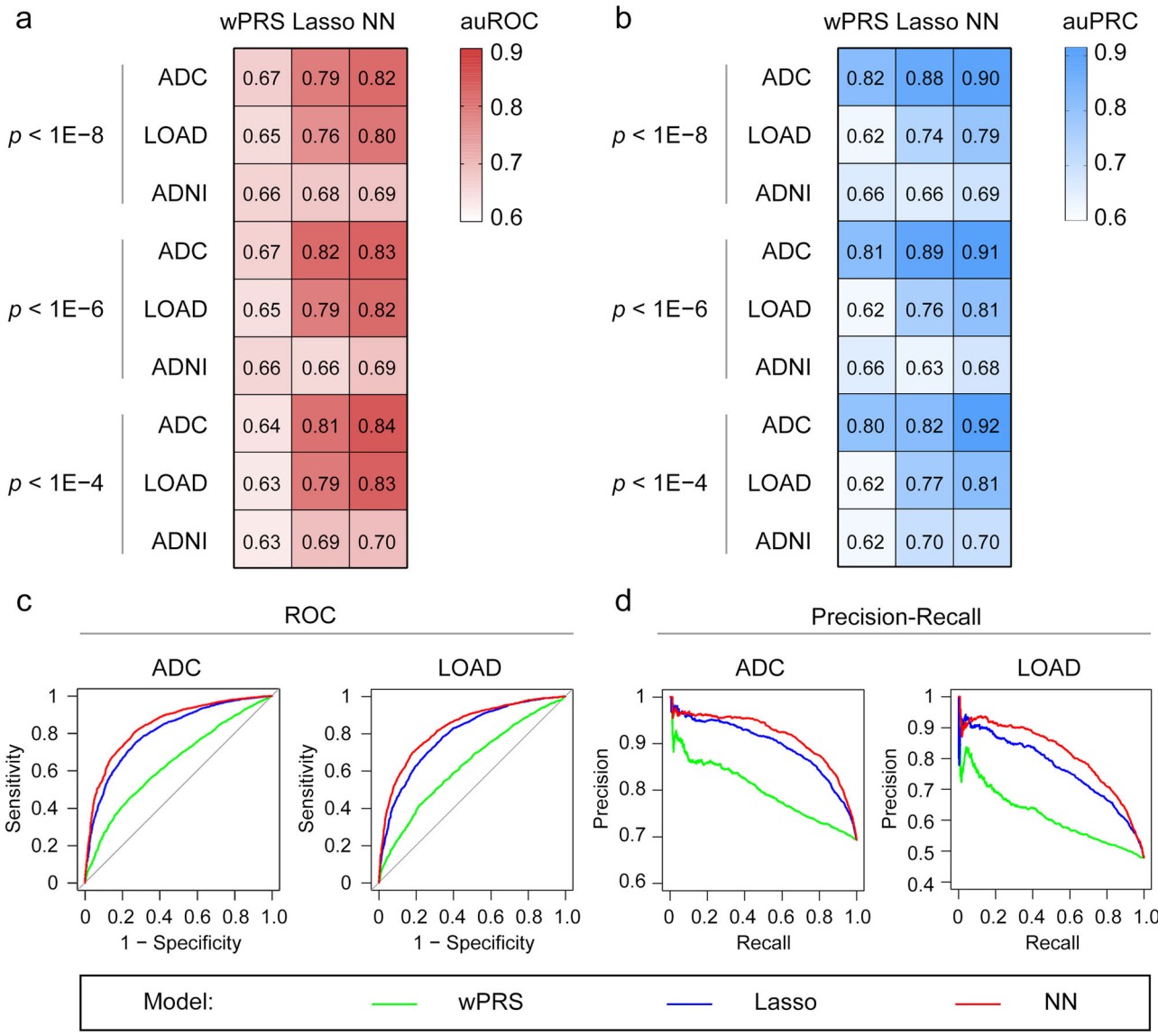

**Fig. 2 Application of the weighted polygenic risk score, lasso, and neural network models for Alzheimer's disease risk classification. a, b** Performance of the wPRS, lasso, and NN models for classifying patients with AD as indicated by (**a**) auROCs and (**b**) auPRCs. The variant pools used for model construction were selected according to the *p*-value cutoffs shown on the left side of each panel. **c, d** Representative plots showing the AD risk classification accuracy of different models constructed using variants with $p < 1E-4$ in individual cohorts. **c** ROC curves and **d** PR curves showing AD risk classification accuracy in different cohorts. AD, Alzheimer's disease; ADC, National Institute on Aging Alzheimer's Disease Center cohort; ADNI, Alzheimer's Disease Neuroimaging Initiative cohort; auPRC, area under the precision–recall curve; auROC, area under the receiver operating characteristic curve; lasso, least absolute shrinkage and selection operator; LOAD, Late Onset Alzheimer's Disease Family Study cohort; NN, neural network; *p*, p-value; PR, precision–recall; ROC, receiver operating characteristic; wPRS, weighted polygenic risk score.

(PI_HAT > 0.90) and reconstruct the models. Consistently, we show that the neural network model outperformed the other two models as suggested by the higher auROC and auPRC values (Supplementary Fig. 5; Supplementary Table 8). Hence, our findings demonstrate the superiority of the neural network model for AD risk classification.

**Effects of confounding factors on Alzheimer's disease risk prediction.** Age and sex are risk factors for AD[61]. Ethnicity also influences AD risk, as the risk effects of specific genetic variants can vary across ethnic groups[62]. Hence, we assess the performance of these polygenic score models in subgroups of people stratified by age, sex, or ethnicity. For weighted PRS models constructed using the GWAS results of European-descent

populations, we find significantly lower accuracy of AD risk classification in people of African-American descent ($n = 713$; auROC $= 0.60$; $p < 0.001$) and Latin-American descent ($n = 604$; auROC $= 0.60$; $p < 0.001$) than in people of European descent ($n = 9940$; auROC $= 0.69$). On the other hand, the neural network model exhibits similar accuracy for classifying AD risk between people of European descent (auROC $= 0.80$) and African-American descent (auROC $= 0.84$) but lower accuracy in people of Latin-American descent (auROC $= 0.77$; $p < 0.05$; Supplementary Fig. 6; Supplementary Table 9). In addition, in people of European descent, we observe similar classification accuracy between males and females (Supplementary Fig. 7; Supplementary Table 10), while older age groups ($\geq 72$ years old) showed higher classification accuracy than younger groups (<72 years old; $p < 0.05$; Supplementary Fig. 8; Supplementary

**Table 1 Model performance for Alzheimer's disease classification.**

| p-value cutoff (number of variants) | Cohort | auROC | | | auPRC | | |
|---|---|---|---|---|---|---|---|
| | | wPRS | Lasso | NN | wPRS | Lasso | NN |
| <1.00E−08 (n = 1799) | ADC | 0.7005 | 0.7894 | 0.8156 | 0.8362 | 0.8820 | 0.9030 |
| | LOAD | 0.6652 | 0.7615 | 0.7995 | 0.6291 | 0.7376 | 0.7909 |
| | ADNI | 0.6835 | 0.6793 | 0.6853 | 0.6690 | 0.6577 | 0.6909 |
| <1.00E−06 (n = 2959) | ADC | 0.7015 | 0.8159 | 0.8273 | 0.8371 | 0.8929 | 0.9095 |
| | LOAD | 0.6657 | 0.7874 | 0.8162 | 0.6291 | 0.7565 | 0.8054 |
| | ADNI | 0.6831 | 0.6555 | 0.6892 | 0.6708 | 0.6340 | 0.6832 |
| <1.00E−04 (n = 8100) | ADC | 0.7003 | 0.8074 | 0.8438 | 0.8377 | 0.8956 | 0.9155 |
| | LOAD | 0.6735 | 0.7939 | 0.8315 | 0.6413 | 0.7696 | 0.8147 |
| | ADNI | 0.6747 | 0.6947 | 0.6956 | 0.6637 | 0.7014 | 0.7016 |

*ADC National Institute on Aging Alzheimer's Disease Centers cohort, ADNI Alzheimer's Disease Neuroimaging Initiative cohort, auPRC area under the precision–recall curve, auROC area under the receiver operating characteristic curve, lasso least absolute shrinkage and selection operator, LOAD Late Onset Alzheimer's Disease Family Study cohort, n number of variants, NN neural network, wPRS weighted polygenic risk score.*

Table 11). Hence, our results suggest that polygenic score models may exhibit variable performance in AD risk classification among people of different ages and ethnic backgrounds.

**Polygenic score models for Alzheimer's disease in the Chinese population.** To further test the performance of these neural network models for classifying AD risk in non-European–descent populations, we apply the models to two Chinese AD cohorts with available WGS data: Chinese WGS cohort 1 (N = 2340 comprising 1116 patients with AD, 309 patients with MCI, and 915 NCs)[18] and Chinese WGS cohort 2 (N = 1077 comprising 356 patients with AD, 68 patients with MCI, and 653 NCs) (Supplementary Table 1)[38]. Notably, the weighted PRS models constructed based on the AD GWAS summary statistics of Jansen et al. show poor classification accuracy for both Chinese WGS cohorts 1 and 2 (auROCs: ~0.50; Supplementary Figs. 9, 10; Supplementary Tables 12–15). Meanwhile, the lasso and neural network models constructed based on three AD cohorts (i.e., ADC, LOAD, and ADNI) classify AD risk in the two Chinese cohorts with moderate accuracy (auROC = 0.63–0.67), although less so than that in the European-descent populations (auROC = 0.72–0.73; Supplementary Fig. 3). Hence, the variants selected based on the AD GWAS summary statistics of Jansen et al. are not representative of AD risk in the Chinese population and are thus unsuitable for constructing polygenic score models for AD in this population.

To obtain variants that are associated with AD in the Chinese population for modeling AD polygenic risk, we gather the AD-associated variants reported from several AD GWASs undertaken across people of different ethnic backgrounds[10,12–15,17,19,63,64], which yielded 216 AD GWAS hits that may contribute to AD (Supplementary Tables 16; Supplementary Data 1). Logistic regression analysis including age, sex, and genomic structure (represented by the top five principal components) as covariates show that 38 of the 216 SNPs were significantly associated with AD in the Chinese population (in either Chinese WGS cohort 1 or 2; Supplementary Data 2, 3). A meta-analysis of the two Chinese cohorts showed that among these 38 SNPs, 33 are significantly associated with AD (meta-p < 0.05; Table 2; Supplementary Data 4) and an additional four SNPs (i.e., rs16824536, rs9271058, rs61732533, and rs111278892) exhibit concordant risk trends in both cohorts (Supplementary Data 4). Thus, we find 37 variants that have been reported in European AD GWASs and are associated with AD in the Chinese population that are useful for modeling AD polygenic risk in the Chinese population.

Using these 37 AD-associated SNPs, we calculate the polygenic scores using the weighted PRS, lasso, and neural network models

in Chinese WGS cohort 1 (see the "Methods" section; Supplementary Data 5). The weighted PRS and lasso models for AD risk classification yielded auROCs of 0.64 and 0.71, respectively (Fig. 3a; Supplementary Fig. 11; Supplementary Table 17), suggesting that the abovementioned variants can be used to classify people at risk of AD in the Chinese population. The modified PRS models (i.e., SBayesR and Winner's curse models) do not show superior performance for AD classification compared to the weighted PRS model (Supplementary Fig. 12). Again, we find that using the variants residing outside the APOE locus is sufficient to distinguish patients with AD from NCs (auROC = 0.61; Supplementary Fig. 11; Supplementary Table 17). Thus, we demonstrated that variants in the non-APOE region contribute to AD pathogenesis, corroborating the findings of other AD polygenic score studies[65,66] and our results in the previous section.

Next, we evaluated whether the neural network model also exhibits better performance for predicting AD in the Chinese population than the weighted PRS and lasso models. Notably, in Chinese WGS cohort 1, the neural network model (auROC = 0.77; auPRC = 0.77) distinguishes patients with AD from NCs more accurately than the weighted PRS (auROC = 0.66; auPRC = 0.71; p < 0.001) and lasso regression models (auROC = 0.71; auPRC = 0.74; p < 0.001) (Fig. 3a; Supplementary Fig. 11; Supplementary Table 17). In addition, the neural network model classifies individuals with MCI with higher accuracy than the other two models (p < 0.01; Supplementary Fig. 11; Supplementary Table 17). To further validate the above results, we examine the accuracy of these models for classifying AD risk in Chinese WGS cohort 2. Notably, the lasso regression (auROC = 0.63; auPRC = 0.51) and neural network models (auROC = 0.63; auPRC = 0.53) perform similarly for AD risk classification and perform slightly better than the weighted PRS model (auROC = 0.62; auPRC = 0.49; Supplementary Fig. 11; Supplementary Table 17). Hence, our analyses in the Chinese population demonstrate the applicability of the neural network model for AD risk classification modeling.

As the selected 37 variants are significantly associated with AD in both European-descent and Chinese populations, they can likely be used to classify AD risk in both populations. Interestingly, by conducting the five-fold cross-validation analysis using these 37 variants separately in European-descent and Chinese populations, the resultant polygenic score models could classify AD risk in both populations (European-descent: auROC = 0.68–0.72; Chinese: auROC = 0.66–0.69) (Supplementary Fig. 13; Supplementary Table 18). In particular, the lasso and neural network models constructed from the 37 variants exhibit

**Table 2 Variants significantly associated with Alzheimer's disease in the two Chinese Alzheimer's disease whole-genome sequencing cohorts.**

| | Variant | EA | Gene | Effect Size (SE) | Meta p-value | I² | EAF (AD/NC) |
|---|---|---|---|---|---|---|---|
| *Variants reported in European-descent populations* | | | | | | | |
| 1 | rs4663105 | C | *BIN1* | 0.190 (0.058) | 1.12E−03 | 0 | 0.506/0.463 |
| 2 | rs6733839 | T | *BIN1* | 0.169 (0.058) | 3.82E−03 | 0 | 0.458/0.418 |
| 3 | rs4504245 | A | *CLNK* | 0.161 (0.078) | 3.80E−02 | 0 | 0.177/0.157 |
| 4 | rs6605556 | G | *HLA-DQA1* | −0.180 (0.088) | 4.03E−02 | 22.49 | 0.107/0.127 |
| 5 | rs9381040 | T | *TREML2* | 0.190 (0.062) | 2.13E−03 | 0 | 0.356/0.333 |
| 6 | rs4236673 | A | *CLU/PTK2B* | −0.147 (0.074) | 4.56E−02 | 0 | 0.190/0.216 |
| 7 | rs11787077 | T | *CLU* | −0.157 (0.072) | 2.85E−02 | 0 | 0.198/0.225 |
| 8 | rs1532278 | T | *CLU* | −0.162 (0.071) | 2.37E−02 | 0 | 0.197/0.226 |
| 9 | rs9331896 | C | *CLU* | −0.143 (0.072) | 4.60E−02 | 0 | 0.198/0.225 |
| 10 | rs867611 | G | *PICALM* | −0.164 (0.060) | 6.14E−03 | 51.73 | 0.353/0.394 |
| 11 | rs561655 | G | *PICALM* | −0.151 (0.058) | 9.85E−03 | 0 | 0.448/0.489 |
| 12 | rs3844143 | C | *PICALM* | −0.142 (0.060) | 1.80E−02 | 34.68 | 0.372/0.410 |
| 13 | rs10792832 | A | *PICALM* | −0.139 (0.060) | 2.01E−02 | 13.18 | 0.364/0.404 |
| 14 | rs3851179 | T | *PICALM* | −0.139 (0.060) | 2.01E−02 | 13.18 | 0.364/0.404 |
| 15 | rs74685827 | G | *SORL1* | 0.188 (0.094) | 4.57E−02 | 0 | 0.115/0.097 |
| 16 | rs11218343 | C | *SORL1* | −0.205 (0.065) | 1.61E−03 | 0 | 0.251/0.291 |
| 17 | rs3752231 | T | *ABCA7* | 0.122 (0.059) | 3.72E−02 | 90.07 | 0.439/0.415 |
| 18 | rs12151021 | G | *ABCA7* | −0.118 (0.058) | 4.30E−02 | 85.93 | 0.475/0.503 |
| 19 | rs3752246 | G | *ABCA7* | 0.128 (0.061) | 3.43E−02 | 87.24 | 0.369/0.345 |
| 20 | rs4147929 | A | *ABCA7* | 0.129 (0.061) | 3.28E−02 | 88.42 | 0.368/0.343 |
| 21 | rs769450 | A | *APOE* | −0.224 (0.074) | 2.42E−03 | 0 | 0.178/0.216 |
| 22 | rs429358 | C | *APOE* | 1.182 (0.086) | 2.72E−43 | 0 | 0.256/0.090 |
| 23 | rs75627662 | T | *APOE* | 0.663 (0.069) | 8.24E−22 | 0 | 0.304/0.186 |
| 24 | rs9304690 | T | *SIGLEC11* | 0.137 (0.070) | 4.89E−02 | 0 | 0.244/0.233 |
| *Variants reported in Asian populations* | | | | | | | |
| 25 | rs11668861 | G | *NECTIN2* | 0.527 (0.066) | 1.71E−15 | 22.76 | 0.321/0.220 |
| 26 | rs6859 | A | *NECTIN2* | 0.455 (0.062) | 1.53E−13 | 0 | 0.408/0.304 |
| 27 | rs3852860 | C | *NECTIN2* | 0.513 (0.065) | 3.60E−15 | 52.36 | 0.332/0.229 |
| 28 | rs71352238 | C | *TOMM40* | 1.045 (0.086) | 9.66E−34 | 0 | 0.230/0.091 |
| 29 | rs157580 | A | *TOMM40* | 0.334 (0.058) | 7.58E−09 | 44.75 | 0.528/0.448 |
| 30 | rs2075650 | G | *TOMM40* | 1.050 (0.086) | 4.11E−34 | 0 | 0.229/0.090 |
| 31 | rs157582 | T | *TOMM40* | 0.685 (0.069) | 3.07E−23 | 0 | 0.315/0.192 |
| 32 | rs439401 | C | *APOE* | 0.422 (0.059) | 6.16E−13 | 62.76 | 0.523/0.431 |
| 33 | rs4420638 | G | *APOC1* | 1.076 (0.080) | 6.41E−41 | 0 | 0.272/0.110 |

The meta-analysis was conducted using a fixed-effects model.
*AD* Alzheimer's disease, *EA* effect allele, *EAF* effect allele frequency, *I²* statistic measuring heterogeneity, *NC* normal control, *SE* standard error.

comparable (or better) performance than the models constructed based on the variants selected by *p*-value thresholds (i.e., 8100, 2959, and 1799 variants) in both the European-descent and Chinese populations (Supplementary Figs. 14, 15). Furthermore, the polygenic score models constructed using the 37 variants based on Chinese data can classify AD risk in the European-descent population (auROC = 0.62–0.65; auPRC = 0.70–0.73), and the models using the same 37 variants based on European-descent data can classify AD risk in the Chinese population (auROC = 0.60–0.67, auPRC = 0.69–0.72; Supplementary Fig. 16; Supplementary Table 19). In addition, the neural network models constructed based on the 37 variants perform significantly better for classifying AD risk than the models constructed with the 216 AD GWAS hits or with other sets of 37 variants randomly selected from the 216 AD GWAS hits (*p* < 0.05; Supplementary Fig. 17). Thus, the polygenic score models based on the 37 variants can be used for modeling and classifying AD risk in both Chinese and European-descent populations.

**Performance of the neural network model for Alzheimer's disease risk classification in the Chinese population.** As the neural network model using the 37 variants show superior performance for classifying AD risk, we examine whether it could stratify individuals with different levels of disease risk. Accordingly, the scores calculated using the neural network model (neural network risk scores hereafter) for individuals in the Chinese cohorts show clear separation between individuals with the lowest and highest scores. We apply a multiple Gaussian fitting model to the neural network risk scores to stratify individuals into low-, medium-, and high-risk groups (see the "Methods" section, Fig. 3c, and Supplementary Fig. 18). Compared with the low-risk group, the medium- and high-risk groups included larger proportions of patients with AD in both Chinese WGS cohorts (e.g., for Chinese WGS cohort 1, patients with AD made up 22.2%, 49.4%, and 70.9%, of the low-, medium- and high-risk groups, respectively) (Fig. 3d; Supplementary Fig. 18; Supplementary Table 20). Furthermore, individuals in the high-risk group have a greater risk of developing AD and MCI than those in the low- or medium-risk group (*p* < 1E−5 and 1E−3 for Chinese WGS cohorts 1 and 2, respectively; Supplementary Table 20). Moreover, in Chinese WGS cohort 1, individuals in the medium-risk group have higher risks of AD (*p* < 2E−16) and MCI (*p* = 1.25E−2) than those in the low-risk group (Supplementary Table 20). Thus, the neural network model can be used to stratify people into subgroups based on their relative risk of developing a disease.

To determine the relevance of the neural network risk scores on clinical outcomes, we examine the association between individuals' scores and their cognitive functioning after controlling for confounding factors (i.e., age, sex, and genomic

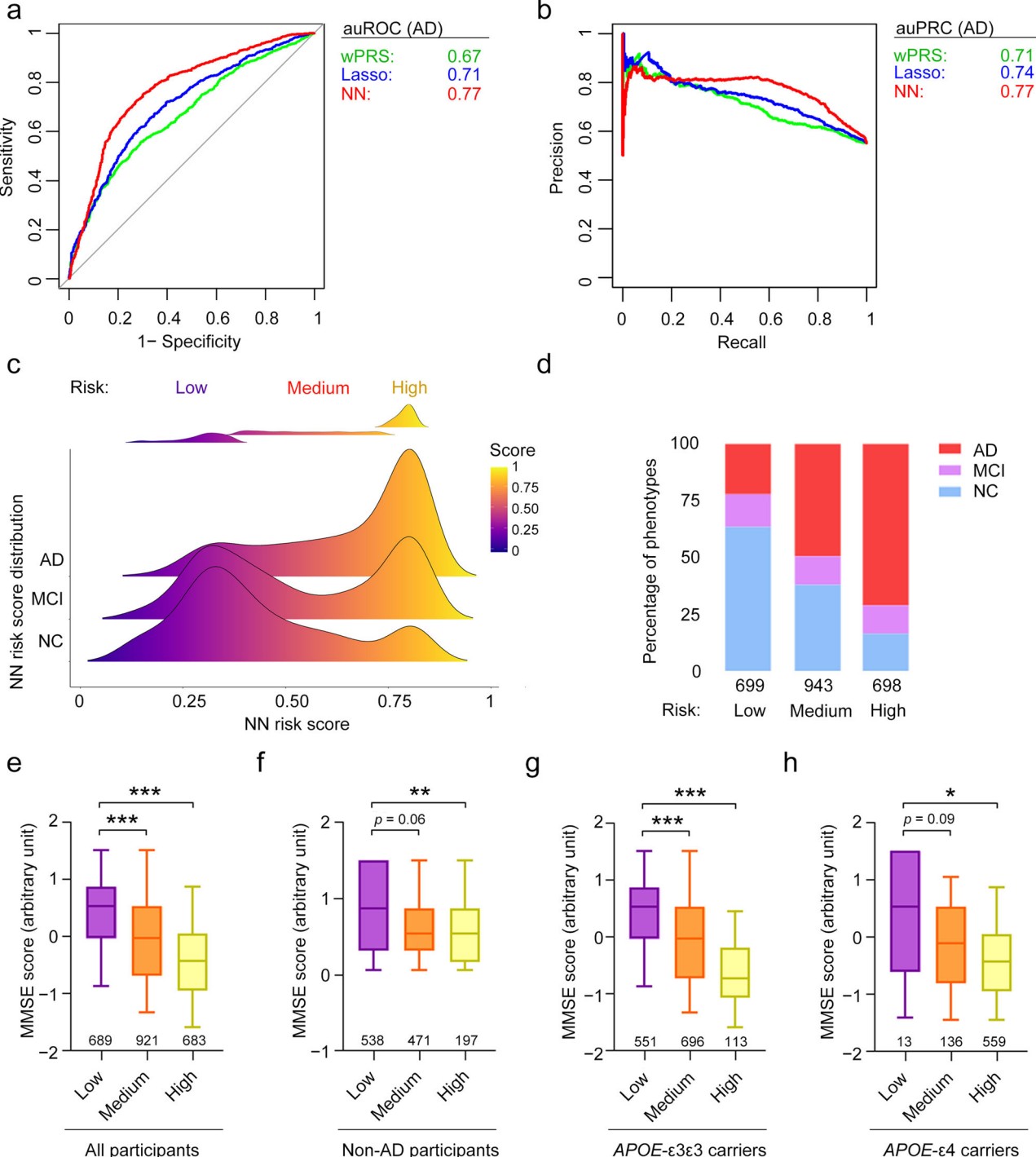

**Fig. 3 Polygenic risk analysis for Alzheimer's disease in the Chinese population. a** ROC and **b** PR curves of the polygenic score classification of patients with AD in Chinese WGS cohort 1. **c** Distribution of polygenic risk scores derived from the NN model for each phenotype group. The definitions of the low-, medium-, and high-risk groups are shown in the upper panel. **d** Percentages of each phenotype group in the low-, medium-, and high-risk groups. **e**–**h** Associations between polygenic risk score and MMSE score in **e** all participants, **f** non-AD participants (i.e., NCs plus patients with MCI), **g** *APOE*-ε3 homozygous carriers, and **h** *APOE*-ε4 carriers. Data are presented as box-and-whisker plots. Boxes indicate the 25th to 75th percentiles, and whiskers indicate the 10th and 90th percentiles. The numbers of individuals in the corresponding group are shown at the bottom of each plot. Robust linear regression model: \*\*\*$p < 0.001$, \*\*$p < 0.01$, \*$p < 0.05$. AD, Alzheimer's disease; auPRC, area under the precision–recall curve; auROC, area under the receiver operating characteristic curve; lasso, least absolute shrinkage and selection operator; MCI, mild cognitive impairment; MMSE, Mini–Mental State Examination; NC, normal control; NN, neural network; *p*, p-values; PR, precision–recall; ROC, receiver operating characteristic; wPRS, weighted polygenic risk score.

structure). Notably, in Chinese WGS cohort 1, the neural network risk scores are significantly associated with cognitive functioning as measured by the Mini–Mental State Examination (MMSE) in all participants ($p < 2E-16$), patients with MCI plus NCs ($p = 3.10E-04$), patients with MCI ($p < 0.05$), APOE-ε3 homozygous participants ($p < 2E-16$), and APOE-ε4 carriers ($p = 2.18E-07$) (Fig. 3e–h; Supplementary Fig. 19; Supplementary Table 21). In addition, in Chinese WGS cohort 2, the neural network risk scores are significantly associated with the Montreal Cognitive Assessment (MoCA) scores of patients with AD plus NCs (Supplementary Fig. 18) as well as those of patients with MCI ($p < 0.05$; Supplementary Fig. 19). Hence, the neural network risk scores calculated herein can predict cognitive functioning in the Chinese population.

**Determination of the pathological mechanisms of Alzheimer's disease according to polygenic scores.** To investigate the mechanisms whereby the identified variants (i.e., SNPs) modulate disease risk, we examine the associations between polygenic risk and AD endophenotypes in Chinese WGS cohort 2[38]. We show that the neural network risk scores are significantly associated with levels of the blood-based ATN biomarkers of classical AD pathology—Aβ, tau phosphorylated at threonine-181 (tau/p-tau181), and NfL—which reflect the progression and severity of AD[67] (Fig. 4a; Supplementary Table 22). Detailed analysis shows that the associations between polygenic scores and plasma biomarker levels are significant in NCs but not in patients with AD, suggesting that the AD risk variants modulate AD-associated pathways independent of disease state (Fig. 4a–d; Supplementary Table 22). Moreover, among all participants, polygenic scores are significantly associated with changes in the volumes of specific brain regions[68,69] including the amygdala ($p = 6.53E-03$), grey matter ($p = 1.21E-02$), and hippocampus ($p = 4.92E-02$) (Supplementary Fig. 20; Supplementary Data 6). Moreover, polygenic scores are significantly associated with white matter hyperintensity, which is a marker of demyelination and axonal loss in the brain[70] ($p = 2.69E-02$; Supplementary Fig. 20; Supplementary Data 6). Hence, our results suggest that AD polygenic risk is associated with known AD biomarkers, particularly in people who have not yet developed AD.

To better understand how AD polygenic risk is associated with endophenotypic changes regardless of disease state, we comprehensively analyze the associations between polygenic scores and the changes in the levels of 1,160 plasma proteins that potentially reflect changes in multiple biological pathways in NCs ($n = 69$). The polygenic scores are significantly associated with the levels of 80 plasma proteins; among these proteins, PLTP (phospholipid transfer protein; $p = 2.67E-03$), which is involved in cholesterol metabolism[71], and CCL19 (chemokine ligand 19; $p = 6.65E-07$), a cytokine involved in inflammation[72], are the most strongly associated with the polygenic scores (Fig. 4e–g; Supplementary Data 7). Specifically, Gene Ontology enrichment analysis suggests that the polygenic scores are associated with plasma proteins involved in TNF-α– and cytokine-related pathways, which are closely related to the immune system[73] (Fig. 4h; Supplementary Data 8, 9). Furthermore, protein–protein interaction network analysis of those plasma proteins involved in cytokine-related pathways again suggests their enriched interaction (enrichment $p < 1E-16$; Fig. 4i; Supplementary Table 32). Together, these results show that AD polygenic risks may modulate immune-associated signaling pathways in the blood.

**Using neural network models to study disease mechanisms.** Given that AD polygenic risks are possibly related to the involvement of multiple biological pathways in disease pathogenesis,

the effects of individual variants on PRSs may partly reflect the contributions of corresponding biological pathways associated with specific genetic variants to the disease. Such effects may not be adequately captured by a single ultimate score derived from polygenic score models but rather by the intermediate outputs of the penultimate layer in neural network models. In our neural network model, the penultimate layer summarizes the polygenic effects of the 37 SNPs into five nodes (Fig. 5a); thus, the outputs from these five nodes may represent distinct genetic risks that affect different biological processes. Accordingly, we find that the outputs from the five nodes are not perfectly correlated (Fig. 5b), suggesting that they contain more information (i.e., polygenic risks) than the final polygenic score. Therefore, we designate each node in the penultimate layer as one module that may account for a distinct biological effect.

To understand the biological effects of the individual modules, we construct a multivariate model that simultaneously incorporates the outcomes of the five modules to determine their associations with individual endophenotypes (i.e., plasma protein levels). Notably, the levels of 336 plasma proteins are significantly associated with the outcomes of the five modules (Supplementary Table 23; Supplementary Data 10). Furthermore, unsupervised clustering analysis shows that these plasma proteins could be classified into four clusters (designated C1–C4) with distinct biological functions (Fig. 5c). For instance, the plasma proteins classify into C1, C3, and C4 are associated with immune pathways; those in C3 and C4 are associated with cell communication; and those in C1 and C4 are associated with TNF-α–related signaling (Fig. 5c).

Accordingly, we hypothesize that the effects of specific risk variants on gene expression regulation—possibly in specific cellular contexts—underlie the observed associations between polygenic risk and plasma protein levels[74]. Thus, to determine whether specific plasma proteins are predominantly expressed in specific blood cell types, we conduct a cell-type enrichment analysis of the plasma proteins in each cluster. Interestingly, the plasma proteins in C1 and C4 are expressed by B cells, those in C2 by erythroblasts and megakaryocytes, and those in C4 by dendritic cells and eosinophils (Fig. 5c; Supplementary Tables 24, 25). Furthermore, protein–protein interaction network analysis reveals that the proteins expressed by B cells are closely interconnected (enrichment $p = 1E-12$; Fig. 5d; Supplementary Tables 26, 27). Specifically, the plasma protein TCL1A (TCL1 family AKT coactivator A), which is uniquely expressed by B cells and associated with B-cell maturation[75], is modulated by polygenic risks; furthermore, its plasma level is altered in patients with AD compared with that in NCs (Fig. 5e, f). Therefore, these results demonstrate that AD polygenic risks are associated with specific biological pathways in a cell-type-specific manner.

To evaluate whether changes in neural network architecture affect the effects of specific risk variants on gene expression regulation, we modify the neural network structure by changing the numbers of nodes in the penultimate layer from five to two, three, or 10 to examine whether the same plasma protein sets can be obtained from the association analysis. First, we find that the neural network risk scores obtain from the modified models are highly correlated ($R^2 > 0.88$; Supplementary Fig. 21a, b). In addition, these modified models recover >80% of the plasma proteins that are previously identified to be associated with the neural network risk scores (i.e., $p < 0.10$; Supplementary Fig. 21c, d). Furthermore, for the neural network model with three nodes in the penultimate layer, the analysis again highlights the associations between polygenic risks and immune-associated signaling pathways such as TNF-α– and cytokine-related pathways (Supplementary Fig. 21e, f). Therefore, these findings

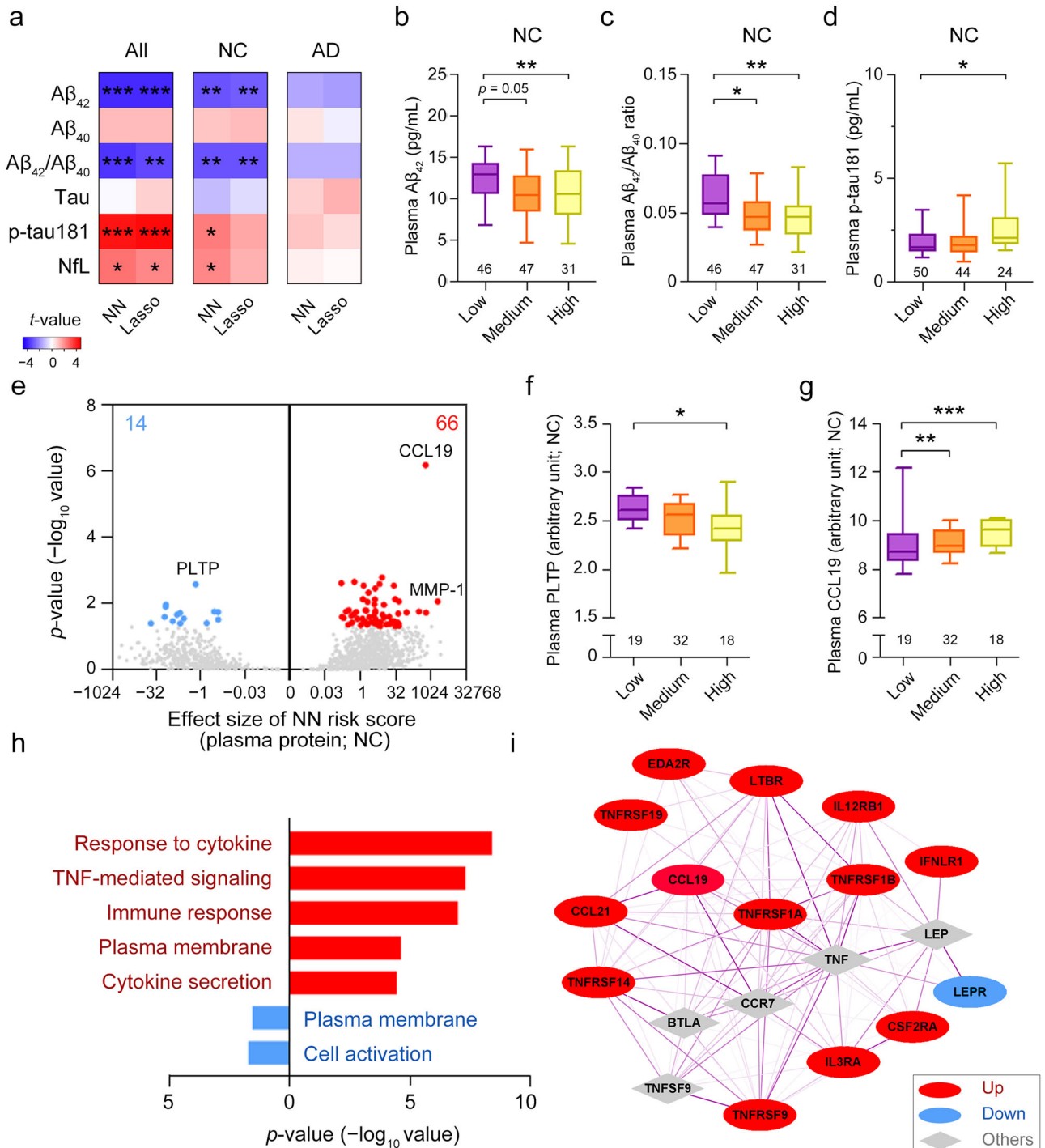

**Fig. 4 Modulatory effects of polygenic risk for Alzheimer's disease on plasma protein biomarkers in normal controls. a** Associations between the polygenic risk scores derived from the corresponding models and the levels of plasma ATN biomarkers (i.e., Aβ$_{42}$, Aβ$_{40}$, Aβ$_{42}$/Aβ$_{40}$ ratio, tau, p-tau181, and NfL) in all participants, NCs, and patients with AD. **b–d** Plasma Aβ$_{42}$ level (**b**), Aβ$_{42}$/Aβ$_{40}$ ratio (**c**), and p-tau181 level (**d**) in NCs stratified according to polygenic risk score group. **e** Volcano plots showing the associations between polygenic risk scores and plasma protein levels obtained from the high-throughput assay. **f**, **g** Levels of the candidate plasma proteins **f** PLTP and **g** CCL19 in NCs stratified according to polygenic risk score group. **h** Overrepresented Gene Ontology terms for plasma proteins associated with polygenic risk scores ($p < 0.05$). **i** Protein–protein interaction network of cytokines associated with polygenic risk scores. The gray nodes are the five proteins most strongly associated with the other nodes. Line color and thickness indicate the interaction strength of the connected nodes (darker and thicker lines denote stronger interactions). **b–d**, **f**, **g** Data are presented as box-and-whisker plots. Boxes indicate the 25th to 75th percentiles, and whiskers indicate the 10th and 90th percentiles; numbers of individuals in the corresponding group are shown at the bottom of each plot. Robust linear regression: \*\*\*$p < 0.001$, \*\*$p < 0.01$, \*$p < 0.05$; robust linear regression model. **e**, **h**, **i** Colors denote plasma proteins or results derived from proteins that were positively (red) or negatively (blue) correlated with polygenic risk scores. Aβ, amyloid-beta; AD, Alzheimer's disease; CCL19, chemokine ligand 19; MCI, mild cognitive impairment; NC, normal control; NfL, neurofilament light polypeptide; NN, neural network; p-tau181, tau phosphorylated at threonine-181; PLTP, phospholipid transfer protein; TNF, tumor necrosis factor.

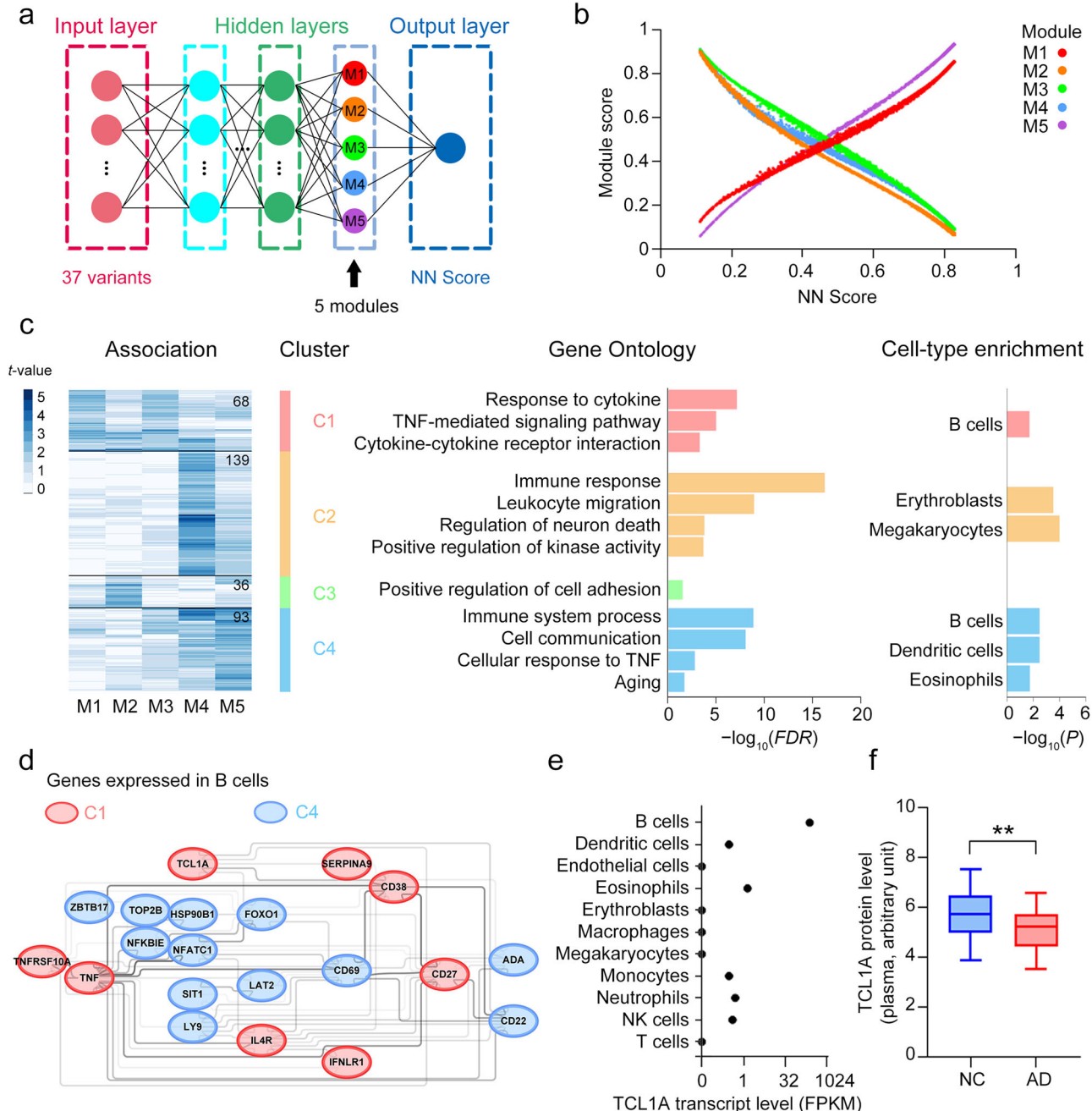

**Fig. 5 Biological pathways modulated by the polygenic risk variants of Alzheimer's disease. a** Diagram showing the calculation of polygenic risk scores using the NN model. The five nodes in the penultimate layer were designated modules M1–M5. **b** Associations between the polygenic risk scores derived from the NN model and the outcomes of the five modules. **c** Heatmap showing the clusters of plasma proteins significantly associated with each module. The proteins formed four clusters (designated C1–C4) with respect to the absolute values of $t$-statistics. The number of proteins in each cluster are indicated in the plot. Representative Gene Ontology terms and cell-type enrichment analysis results are displayed in the center and right panels, respectively. **d** Protein–protein interaction network of proteins expressed by B cells. Colors denote proteins from C1 (red) and C4 (blue). **e** Cell-type-specific expression of TCL1A. (f) Plasma levels of TCL1A protein in NCs ($n = 69$) and patients with AD ($n = 97$). Data are presented as box-and-whisker plots. Boxes indicate the 25th to 75th percentiles, and whiskers indicate the 10th and 90th percentiles; numbers indicate the numbers of individuals in the corresponding group. Robust linear regression: ** $p < 0.01$. AD, Alzheimer's disease; FDR, false discovery rate; FPKM, fragments per kilobase per million mapped fragments; NC, normal control; NK, natural killer; NN, neural network; $p$, p-values; TCL1A, TCL1 family AKT coactivator A; TNF, tumor necrosis factor.

further strengthen our conclusions on the association of AD polygenic risk with immune-associated pathways.

**Using neural network models to stratify people at risk of developing Alzheimer's disease.** The intermediate outputs from

the neural network model capture polygenic risks that correspond to multiple biological pathways implicated in AD pathogenesis. Therefore, it is of interest to examine whether this model can stratify people into subgroups based on the polygenic risks estimated by those intermediate outputs. Accordingly, we subject the outputs from the penultimate layer of the neural network model

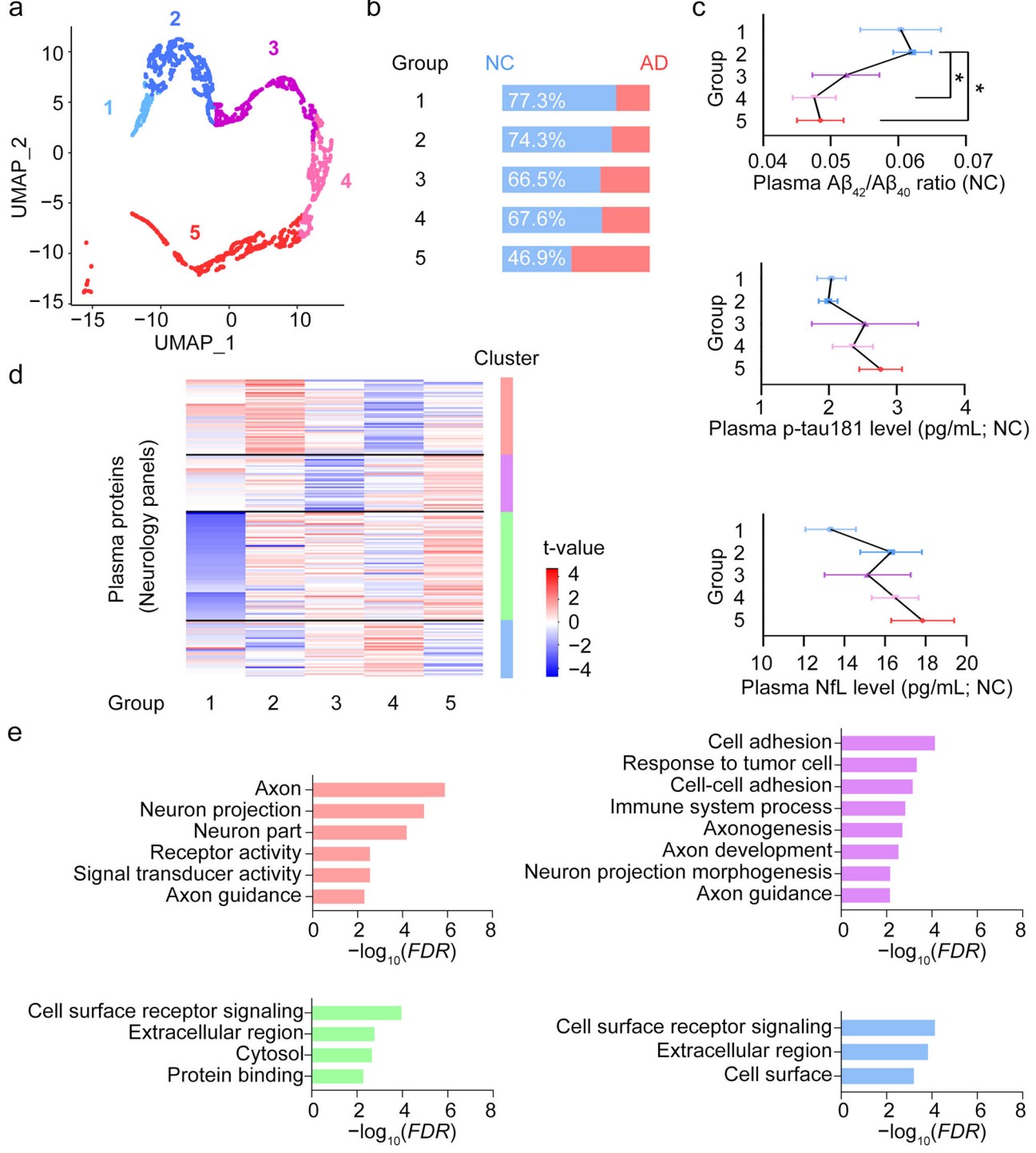

**Fig. 6 Stratification of individuals by polygenic risk score from neural network models. a** *K*-means clustering of the individuals in the Chinese AD WGS cohort 2 dataset according to the five sub-scores from the NN model. **b** Proportion of NCs in each group. **c** Levels of plasma ATN biomarkers in individual groups (*n* = 16, 41, 22, 29, and 34 individuals in Groups 1–5, respectively). Data are presented as mean ± SEM and analyzed using one-way ANOVA followed by Bonferroni's *post hoc* test. *$p < 0.05$. **d** Heatmap of association *t*-values between plasma protein levels detected by two neurology panels and individual groups. According to their *t*-values, proteins were divided into four clusters using the *k*-means method (number of proteins in each cluster = 46, 35, 67, 35, from top to bottom, accordingly). **e** Pathway and Gene Ontology enrichment analysis results for proteins in each cluster. Aβ, amyloid-beta; AD, Alzheimer's disease; ATN, amyloid-beta, tau, and neurofilament light polypeptide; FDR, false discovery rate; NC, normal control; NfL, neurofilament light polypeptide; NN, neural network; p-tau181, tau phosphorylated at threonine-181; SEM, standard error of the mean; UMAP, Uniform Manifold Approximation and Projection.

to unsupervised clustering analysis and then subcluster the participants from Chinese AD WGS cohort 2 into five groups. Of note, the NCs in Groups 4 and 5 showed lower plasma levels of Aβ ($p < 0.05$) and an increased trend of plasma p-tau181 and NfL levels compared with the NCs in Groups 1–3 (Fig. 6a–c). Further association analysis identifies four clusters of plasma proteins that exhibited altered expression patterns among the five groups of individuals. Gene Ontology and pathway analysis reveal that the altered pathways included axon ($p = 1.30E-06$), neuron projection ($p = 1.10E-05$), and receptor activity ($p = 2.80E-03$) (Fig. 6d). Thus, the neural network model can be used to classify AD risk for individuals as well as provide insights into the disease mechanisms based on their polygenic risk information.

**Modeling of disease risk by polygenic score**. To identify which variants play critical roles in our neural network model for AD risk classification, we prioritize the variants according to their biological properties and use partial correlation analysis to estimate their relative contributions to the final neural network risk scores (Supplementary Fig. 22). Interestingly, the variants involved in the regulation of biological functions (e.g., residing in coding regions or transcription factor binding regions) showed greater contributions to the obtained polygenic scores (Supplementary Fig. 22a). For instance, coding variant rs429358, which encodes *APOE*-ε4 and is one of the most well-accepted AD genetic risk factors, is significantly correlated with the obtained risk scores (Spearman's $rho = 0.24$, $p < 0.001$; Supplementary Fig. 22a). Meanwhile, the noncoding variant rs439401, identified as an AD risk factor that exerts a risk effect independent of the *APOE*-ε4 genotype[76], is also significantly associated with the obtained risk scores (Spearman's $rho = 0.05$, $p < 0.001$; Supplementary Fig. 22a). Of note, rs439401 resides in the regulatory region and occupies the transcription factor-binding regions, which may influence the expression of specific genes (Supplementary Fig. 22b, c). Furthermore, our genotype–expression analysis reveals the association between rs439401 and altered *APOE* expression in skin tissues (Supplementary Fig. 22d). Meanwhile, brain single-cell ATAC-seq data suggests that rs439401 resides in the open chromatin regions of specific brain cells, further supporting the roles of rs439401 in regulating *APOE* gene in the brain (Supplementary Fig. 22e)[56]. Thus, variants with specific biological functions might have a stronger effect on modulating disease risk, making them more informative for classifying disease risk.

## Discussion

Here, we present the first deep learning-based polygenic score analysis for AD to the best of our knowledge. We evaluate the performance of weighted PRS, lasso, and neural network models for predicting AD risk based on genetic information and show that the deep learning model classifies disease risk more accurately than the weighted PRS and lasso models. When classifying clinically diagnosed AD patients, the best auROC our neural network model achieved is 0.84, which is higher than other recently reported results based on the weighted PRS model (auROC = 0.74)[30]. Meanwhile, by associating the risk scores (as well as the outputs of the hidden layers) from the neural network model with the disease-associated endophenotypes (e.g., cognitive function and the plasma proteome), we identify how AD polygenic risk may be correlated with pathophysiological changes in individual patients. Furthermore, we show that deep learning methods can stratify people at risk of developing diseases into subgroups according to their polygenic risks (Fig. 6)[77]. Thus, this study highlights the potential of using deep learning methods to investigate disease mechanisms and stratify at-risk people into subgroups, thereby paving the way to develop precision medicine for early disease intervention.

While the neural network model can be used for polygenic risk analysis of AD, there is room to improve the model's performance. First, incorporating more variants into the classification model may better capture the genetic signatures that contribute to the disease, thereby increasing the accuracy of disease classification. Meanwhile, misdiagnoses and misclassification of patients (or NCs) may affect the accuracy of the model; this can be improved by better defining the classification of the disease with disease biomarkers such as brain amyloid load and levels of fluid biomarkers (e.g., Aβ and p-tau181) for AD[64,78]. As most genetic and polygenic risk analyses are performed in European-descent populations, it would also be beneficial to conduct more studies in non-European–descent populations to better understand the disease-associated genetic risks and develop customized polygenic score models for early risk prediction in distinct ethnic populations[79].

Disease-associated variants may modify disease risk by affecting specific biological processes. Notably, our results suggest that functional variants are likely to contribute more to the polygenic risk model when modeling the disease risk (Supplementary Fig. 22). Thus, incorporating the biological properties of variants may enhance the model's accuracy for classifying AD risk. Accordingly, we construct a graph neural network model by integrating allele dosage, annotated functions, and the LD of variants, which exhibited superior classification accuracy compared with the weighted PRS model ($p < 0.001$; Supplementary Fig. 23). Thus, it is critical to conduct further research on the interpretability of deep learning models[80,81] and the usefulness of different types of deep learning models (e.g., the graph neural network model) for modeling disease risk to gain a comprehensive understanding of disease mechanisms and develop more accurate models for disease risk forecasting using genetic data.

Taken together, our results suggest the utility of deep learning methods for predicting disease risk and stratifying people at risk of developing diseases into subgroups as well as their potential applications in uncovering disease mechanisms. Further studies are required to explore the utility of these methods for predicting disease risk at a population scale as well as their potential applications in disease mechanism studies and therapeutic development.

## Data availability

All data associated with this study are in the main text and the Supplementary Information or Supplementary Data. Source data for the figures are available as tables in Supplementary Information or Supplementary Data. Supplementary Data 1–10 can be found in the Supplementary Data file as separate spreadsheets. The genotype data used in the study for variant selection can be accessed in the corresponding sources: the National Institute on Aging–Late Onset Alzheimer's Disease Family Study cohort (LOAD) raw data can be accessed in the database of Genotypes and Phenotypes (dbGaP) at phs000168.v2.p2; the Alzheimer's Disease Genetics Consortium (ADGC) Genome Wide Association Study–NIA Alzheimer's Disease Centers cohort (ADC) raw data can be accessed in the dbGaP at phs000372.v1.p1; and the Alzheimer's Disease Neuroimaging Initiative cohort (ADNI) dataset can be accessed in the ADNI database (https://adni.loni.usc.edu/). The genetic and Alzheimer's disease-associated endophenotypic data analysis results are provided in the Supplementary Information. For data from the Chinese population, the consent form signed by individual participants states that the research content will be kept private under the supervision of the hospital and research team. Therefore, these data will be made available and shared only in the context of a formal collaboration; applications for data sharing and project collaboration will be processed and reviewed by a Review Panel hosted at the Hong Kong University of Science and Technology. Researchers may contact sklneurosci@ust.hk for further details on project collaboration and the sharing of the data from this study.

## Code availability

The code for the neural network for polygenic score analysis (NNP) together with the dummy datasets has been deposited at GitHub (https://github.com/xzhouai/NNP; https://doi.org/10.5281/zenodo.7566919)[82].

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

## Acknowledgements

We thank Pauline Kwan, Hazel Mok, Dr. Phillip Y.C. Chan, Choi Ying Ling, and Bonnie W. Wong for coordinating the collection of clinical data. We also thank Ka Chun Lok, Cara Wing Si Kwong, Tiffany Tze Wing Mak, Yan Ma, and Saijuan Liu for their excellent technical assistance as well as other members of the Ip Laboratory for many helpful discussions. This work was supported in part by the National Key R&D Program of China (2021YFE0203000); the Research Grants Council of Hong Kong (the General Research Fund [16103122], the Theme-Based Research Scheme [T13-607/12R], and the Collaborative Research Fund [C6027-19GF]); the Areas of Excellence Scheme of the University Grants Committee (AoE/M-604/16); the Innovation and Technology Commission (ITCPD/17-9); the InnoHK; the Guangdong Provincial Fund for Basic and Applied Basic Research (2019B1515130004); the NSFC-RGC Joint Research Scheme (32061160472); the Guangdong Provincial Key S&T Program (2018B030336001); the Shenzhen Knowledge Innovation Program (JCYJ20180507183642005 and JCYJ20200109115631248); and the Guangdong–Hong Kong–Macao Greater Bay Area Center for Brain Science and Brain-Inspired Intelligence Fund (2019001 and 2019003). We obtained a portion of the data used in the preparation of this article from the ADNI database (https://adni.loni.usc.edu/); as such, the investigators within the ADNI contributed to the design and implementation of that initiative and/or provided data but did not participate in the analysis or writing of this report. For the ADNI dataset, data collection and sharing for this project were funded by the ADNI (NIH grant number: U01-AG024904) and the DoD ADNI (Department of Defense award number: W81XWH12-2-0012). The ADNI is funded by the NIA and the National Institute of Biomedical Imaging and Bioengineering as well as generous contributions from the following organizations: AbbVie, the Alzheimer's Association, the Alzheimer's Drug Discovery Foundation, Araclon Biotech, BioClinica Inc., Biogen, Bristol Myers Squibb Company, CereSpir Inc., Cogstate, Eisai Inc., Elan Pharmaceuticals Inc., Eli Lilly and Company, EuroImmun, F. Hoffmann–La Roche Ltd. and its affiliated company Genentech Inc., Fujirebio, GE Healthcare, IXICO Ltd., Janssen Alzheimer Immunotherapy Research & Development LLC, Johnson & Johnson Pharmaceutical Research & Development LLC, Lumosity, Lundbeck, Merck & Co. Inc., Meso Scale Diagnostics LLC, NeuroRx Research, Neurotrack Technologies, Novartis 20 Pharmaceuticals Corporation, Pfizer Inc., Piramal Imaging, Servier, Takeda Pharmaceutical Company, and Transition Therapeutics. The Canadian Institutes of Health Research provides funding to support ADNI clinical sites in Canada. Private-sector contributions are facilitated by the Foundation for the National Institutes of Health (www.fnih.org). The grantee organization is the Northern California Institute for Research and Education, and the study is coordinated by the Alzheimer's Therapeutic Research Institute at the University of Southern California. The data generated by the ADNI are disseminated by the Laboratory for Neuro Imaging at the University of Southern California. For the ADGC Genome Wide Association Study–NIA Alzheimer's Disease Centers cohort (i.e., the ADC dataset), funding support for the ADGC was provided through the NIA Division of Neuroscience (grant number: U01-AG032984). For the NIA–Late Onset Alzheimer's Disease Family Study (i.e., the LOAD dataset), funding support for "the Genetic Consortium for Late Onset Alzheimer's Disease" was provided through the NIA Division of Neuroscience. The Genetic Consortium for Late Onset Alzheimer's Disease includes a genome-wide association study funded as part of the NIA Division of Neuroscience. The Genetic Consortium for Late Onset Alzheimer's Disease assisted with phenotype harmonization and genotype cleaning as well as general study coordination. A complete listing of ADNI investigators can be found at https://adni.loni.usc.edu/wp-content/uploads/how_to_apply/ADNI_Acknowledgement_List.pdf.

## Author contributions

X.Z., K.Y.M., A.K.F., J.H., L.C., and N.Y.I. conceived of the project; Y.C., F.C.I., T.Y., V.C.M., T.C.K., and Q.G. organized patient recruitment and sample collection; Y.Z., S.M., R.M.L., and E.P.T. performed the experiment; X.Z., G.L., and J.C. set up the data-processing pipelines; X.Z., Y.J., H.C., G.L., H.Z., J.C., M.S., K.Y.M., J.H., L.C, A.K.F., and N.Y.I. analyzed the data; X.Z., A.K.F., L.C., and N.Y.I. wrote the manuscript; the Alzheimer's Disease Neuroimaging Initiative contributed part of the study data.

## Competing interests

The authors declare no competing interests.

## Additional information

[1]Division of Life Science, State Key Laboratory of Molecular Neuroscience, Molecular Neuroscience Center, The Hong Kong University of Science and Technology, Clear Water Bay, Kowloon, Hong Kong, China. [2]Hong Kong Center for Neurodegenerative Diseases, Hong Kong Science Park, Hong Kong, China. [3]Guangdong Provincial Key Laboratory of Brain Science, Disease and Drug Development, HKUST Shenzhen Research Institute, Shenzhen–Hong Kong Institute of Brain Science, Shenzhen, Guangdong 518057, China. [4]Chinese Academy of Sciences Key Laboratory of Brain Connectome and Manipulation, Shenzhen Key Laboratory of Translational Research for Brain Diseases, The Brain Cognition and Brain Disease Institute, Shenzhen Institute of Advanced Technology, Chinese Academy of Sciences, Shenzhen–Hong Kong Institute of Brain Science-Shenzhen Fundamental Research Institutions, Shenzhen, Guangdong 518055, China. [5]Department of Computer Science and Engineering, The Hong Kong University of Science and Technology, Clear Water Bay, Kowloon, Hong Kong, China. [6]Gerald Choa Neuroscience Centre, Lui Che Woo Institute of Innovative Medicine, Therese Pei Fong Chow Research Centre for Prevention of Dementia, Division of Neurology, Department of Medicine and Therapeutics, The Chinese University of Hong Kong, Shatin, Hong Kong, China. [7]Therese Pei Fong Chow Research Centre for Prevention of Dementia, Division of Geriatrics, Department of Medicine and Therapeutics, The Chinese University of Hong Kong, Shatin, Hong Kong, China. [8]Department of Gerontology, Shanghai Jiao Tong University Affiliated Sixth People's Hospital, Shanghai 200233, China. [9]Department of Neurodegenerative Disease, UCL Queen Square Institute of Neurology, London, UK. [10]UK Dementia Research Institute at UCL, London, UK. [11]HKUST Jockey Club Institute for Advanced Study, The Hong Kong University of Science and Technology, Clear Water Bay, Kowloon, Hong Kong, China. *A list of authors and their affiliations appears at the end of the paper. ✉email: boip@ust.hk

## Alzheimer's Disease Neuroimaging Initiative

Michael W. Weiner[12], Paul Aisen[13], Ronald Petersen[14], Clifford R. Jack[14], William Jagust[15], John Q. Trojanowski[16], Arthur W. Toga[17], Laurel Beckett[18], Robert C. Green[19], Andrew J. Saykin[20], John Morris[21], Leslie M. Shaw[16], Zaven Khachaturian[18,22], Greg Sorensen[23], Lew Kuller[24], Marcus Raichle[21], Steven Paul[25], Peter Davies[26], Howard Fillit[27], Franz Hefti[28], David Holtzman[21], Marek M. Mesulam[29], William Potter[30], Peter Snyder[31], Adam Schwartz[32], Tom Montine[33], Ronald G. Thomas[33], Michael Donohue[33], Sarah Walter[33], Devon Gessert[33], Tamie Sather[33], Gus Jiminez[33], Danielle Harvey[18], Matthew Bernstein[14], Paul Thompson[34], Norbert Schuff[12,18], Bret Borowski[14], Jeff Gunter[14], Matt Senjem[14], Prashanthi Vemuri[14], David Jones[14], Kejal Kantarci[14], Chad Ward[14], Robert A. Koeppe[35], Norm Foster[36], Eric M. Reiman[37], Kewei Chen[37], Chet Mathis[27], Susan Landau[15], Nigel J. Cairns[21], Erin Householder[21], Lisa Taylor-Reinwald[21], Virginia Lee[16], Magdalena Korecka[16], Michal Figurski[16], Karen Crawford[17], Scott Neu[17], Tatiana M. Foroud[20], Steven G. Potkin[38], Li Shen[20], Kelley Faber[20], Sungeun Kim[20], Kwangsik Nho[20], Leon Thal[13], Neil Buckholtz[39], Marylyn Albert[40], Richard Frank[41], John Hsiao[39], Jeffrey Kaye[42], Joseph Quinn[42], Betty Lind[42], Raina Carter[42], Sara Dolen[42], Lon S. Schneider[17], Sonia Pawluczyk[17], Mauricio Beccera[17], Liberty Teodoro[17], Bryan M. Spann[17], James Brewer[13], Helen Vanderswag[13], Adam Fleisher[13,37], Judith L. Heidebrink[35], Joanne L. Lord[35], Sara S. Mason[14], Colleen S. Albers[14], David Knopman[14], Kris Johnson[14], Rachelle S. Doody[43], Javier Villanueva-Meyer[43], Munir Chowdhury[43], Susan Rountree[43], Mimi Dang[43], Yaakov Stern[43], Lawrence S. Honig[43], Karen L. Bell[43], Beau Ances[21], Maria Carroll[21], Sue Leon[21], Mark A. Mintun[21], Stacy Schneider[21], Angela Oliver[21], Daniel Marson[44], Randall Griffith[44], David Clark[44], David Geldmacher[44], John Brockington[44], Erik Roberson[44], Hillel Grossman[45], Effie Mitsis[45], Leyla de Toledo-Morrell[46], Raj C. Shah[46], Ranjan Duara[47], Daniel Varon[47], Maria T. Greig[47], Peggy Roberts[47], Chiadi Onyike[40], Daniel D'Agostino[40], Stephanie Kielb[40], James E. Galvin[48], Brittany Cerbone[48], Christina A. Michel[48],

Henry Rusinek[48], Mony J. de Leon[48], Lidia Glodzik[48], Susan De Santi[48], P. Murali Doraiswamy[49], Jeffrey R. Petrella[49], Terence Z. Wong[49], Steven E. Arnold[16], Jason H. Karlawish[16], David Wolk[16], Charles D. Smith[50], Greg Jicha[50], Peter Hardy[50], Partha Sinha[50], Elizabeth Oates[50], Gary Conrad[50], Oscar L. Lopez[24], MaryAnn Oakley[24], Donna M. Simpson[40], Anton P. Porsteinsson[51], Bonnie S. Goldstein[52], Kim Martin[52], Kelly M. Makino[52], M. Saleem Ismail[52], Connie Brand[52], Ruth A. Mulnard[38], Gaby Thai[38], Catherine McAdams-Ortiz[38], Kyle Womack[52], Dana Mathews[52], Mary Quiceno[52], Ramon Diaz-Arrastia[52], Richard King[52], Myron Weiner[52], Kristen Martin-Cook[52], Michael DeVous[52], Allan I. Levey[53], James J. Lah[53], Janet S. Cellar[53], Jeffrey M. Burns[54], Heather S. Anderson[54], Russell H. Swerdlow[54], Liana Apostolova[34], Kathleen Tingus[34], Ellen Woo[34], Daniel H. S. Silverman[34], Po H. Lu[34], George Bartzokis[34], Neill R. Graff-Radford[55], Francine Parfitt[55], Tracy Kendall[55], Heather Johnson[55], Martin R. Farlow[20], Ann Marie Hake[20], Brandy R. Matthews[20], Scott Herring[20], Cynthia Hunt[20], Christopher H. van Dyck[56], Richard E. Carson[56], Martha G. MacAvoy[56], Howard Chertkow[57], Howard Bergman[57], Chris Hosein[57], Ging-Yuek Robin Hsiung[58], Howard Feldman[58], Benita Mudge[58], Michele Assaly[58], Charles Bernick[59], Donna Munic[59], Andrew Kertesz[60], John Rogers[60], Dick Trost[60], Diana Kerwin[29], Kristine Lipowski[29], Chuang-Kuo Wu[29], Nancy Johnson[29], Carl Sadowsky[61], Walter Martinez[61], Teresa Villena[61], Raymond Scott Turner[62], Kathleen Johnson[62], Brigid Reynolds[62], Reisa A. Sperling[19], Keith A. Johnson[19], Gad Marshall[19], Meghan Frey[19], Barton Lane[19], Allyson Rosen[19], Jared Tinklenberg[19], Marwan N. Sabbagh[63], Christine M. Belden[63], Sandra A. Jacobson[63], Sherye A. Sirrel[63], Neil Kowall[63], Ronald Killiany[64], Andrew E. Budson[64], Alexander Norbash[64], Patricia Lynn Johnson[64], Joanne Allard[65], Alan Lerner[66], Paula Ogrocki[66], Leon Hudson[66], Evan Fletcher[18], Owen Carmichael[18], John Olichney[18], Charles DeCarli[18], Smita Kittur[67], Michael Borrie[68], T-Y. Lee[68], Rob Bartha[68], Sterling Johnson[69], Sanjay Asthana[69], Cynthia M. Carlsson[69], Adrian Preda[34], Dana Nguyen[34], Pierre Tariot[36], Stephanie Reeder[36], Vernice Bates[70], Horacio Capote[70], Michelle Rainka[70], Douglas W. Scharre[71], Maria Kataki[71], Anahita Adeli[71], Earl A. Zimmerman[72], Dzintra Celmins[72], Alice D. Brown[72], Godfrey D. Pearlson[73], Karen Blank[73], Karen Anderson[73], Robert B. Santulli[74], Tamar J. Kitzmiller[74], Eben S. Schwartz[74], Kaycee M. Sink[75], Jeff D. Williamson[75], Pradeep Garg[75], Franklin Watkins[75], Brian R. Ott[76], Henry Querfurth[76], Geoffrey Tremont[76], Stephen Salloway[77], Paul Malloy[77], Stephen Correia[77], Howard J. Rosen[12], Bruce L. Miller[12], Jacobo Mintzer[78], Kenneth Spicer[78], David Bachman[78], Stephen Pasternak[60], Irina Rachinsky[60], Dick Drost[60], Nunzio Pomara[79], Raymundo Hernando[79], Antero Sarrael[79], Susan K. Schultz[80], Laura L. Boles Ponto[80], Hyungsub Shim[80], Karen Elizabeth Smith[80], Norman Relkin[25], Gloria Chaing[25], Lisa Raudin[22,25], Amanda Smith[81], Kristin Fargher[81], Balebail Ashok Raj[81], Thomas Neylan[12], Jordan Grafman[29], Melissa Davis[13], Rosemary Morrison[13], Jacqueline Hayes[12], Shannon Finley[12], Karl Friedl[82], Debra Fleischman[46], Konstantinos Arfanakis[46], Olga James[49], Dino Massoglia[78], J. Jay Fruehling[69], Sandra Harding[69], Elaine R. Peskind[33], Eric C. Petrie[71], Gail Li[71], Jerome A. Yesavage[83], Joy L. Taylor[83] & Ansgar J. Furst[83]

[12]UC San Francisco, San Francisco, CA 94143, USA. [13]UC San Diego, San Diego, CA 92093, USA. [14]Mayo Clinic, Rochester, NY 14603, USA. [15]UC Berkeley, Berkeley, CA 94720, USA. [16]UPenn, Philadelphia, PA 9104, USA. [17]USC, Los Angeles, CA 90089, USA. [18]UC Davis, Davis, CA 95616, USA. [19]Brigham and Women's Hospital/Harvard Medical School, Boston, MA 02115, USA. [20]Indiana University, Bloomington, IN 47405, USA. [21]Washington University in St Louis, St Louis, MI 63130, USA. [22]Prevent Alzheimer's Disease 2020, Rockville, MD 20850, USA. [23]Siemens, Munich 80333, Germany. [24]University of Pittsburgh, Pittsburgh, PA 15260, USA. [25]Weill Cornell Medical College, Cornell University, New York City, NY 10065, USA. [26]Albert Einstein College of Medicine of Yeshiva University, Bronx, NY 10461, USA. [27]AD Drug Discovery Foundation, New York City, NY 10019, USA. [28]Acumen Pharmaceuticals, Livermore, CA 94551, USA. [29]Northwestern University, Evanston and Chicago, IL 60208, USA. [30]National Institute of Mental Health, Rockville, MD 20852, USA. [31]Brown University, Providence, RI 02912, USA. [32]Eli Lilly, Indianapolis, IN 46225, USA. [33]University of Washington, Seattle, WA 98195, USA. [34]UCLA, Los Angeles, CA 90095, USA. [35]University of Michigan, Ann Arbor, MI 48109, USA. [36]University of Utah, Salt Lake City, UT 84112, USA. [37]Banner Alzheimer's Institute, Phoenix, AZ 85006, USA. [38]UC Irvine, Irvine, CA 92697, USA. [39]National Institute on Aging, Bethesda, MD 20892, USA. [40]Johns Hopkins University, Baltimore, MD 21218, USA. [41]Richard Frank Consulting, Washington 20001, USA. [42]Oregon Health and Science University, Portland, OR 97239, USA. [43]Baylor College of Medicine, Houston, TX 77030, USA. [44]University of Alabama, Birmingham, AL 35233, USA. [45]Mount Sinai School of Medicine, New York City, NY 10029, USA. [46]Rush

University Medical Center, Chicago, IL 60612, USA. [47]Wien Center, Miami, FL 33140, USA. [48]New York University, New York City, NY 10003, USA. [49]Duke University Medical Center, Durham, NC 27710, USA. [50]University of Kentucky, Lexington, KY 0506, USA. [51]University of Rochester Medical Center, Rochester, NY 14642, USA. [52]University of Texas Southwestern Medical School, Dallas, TX 75390, USA. [53]Emory University, Atlanta, GA 30322, USA. [54]University of Kansas Medical Center, Kansas City, KS 66103, USA. [55]Mayo Clinic, Jacksonville, FL 32224, USA. [56]Yale University School of Medicine, New Haven, CT 06510, USA. [57]McGill University/Montreal-Jewish General Hospital, Montreal, QC H3T 1E2, Canada. [58]University of British Columbia Clinic for AD & Related Disorders, Vancouver, BC V6T 1Z3, Canada. [59]Cleveland Clinic Lou Ruvo Center for Brain Health, Las Vegas, NV 89106, USA. [60]St Joseph's Health Care, London, ON N6A 4V2, Canada. [61]Palm Beach Neurology Premiere Research Institute, Miami, FL 33407, USA. [62]Georgetown University Medical Center, Washington, DC 20007, USA. [63]Banner Sun Health Research Institute, Sun City, AZ 85351, USA. [64]Boston University, Boston, MA 02215, USA. [65]Howard University, Washington, DC 20059, USA. [66]Case Western Reserve University, Cleveland, OH 20002, USA. [67]Neurological Care of CNY, Liverpool, NY 13088, USA. [68]Parkwood Hospital, London, ON N6C 0A7, Canada. [69]University of Wisconsin, Madison, WI 53706, USA. [70]Dent Neurologic Institute, Amherst, NY 14226, USA. [71]Ohio State University, Columbus, OH 43210, USA. [72]Albany Medical College, Albany, NY 12208, USA. [73]Hartford Hospital, Olin Neuropsychiatry Research Center, Hartford, CT 06114, USA. [74]Dartmouth- Hitchcock Medical Center, Lebanon, NH 03766, USA. [75]Wake Forest University Health Sciences, Winston-Salem, NC 27157, USA. [76]Rhode Island Hospital, Providence, RI 02903, USA. [77]Butler Hospital, Providence, RI 02906, USA. [78]Medical University South Carolina, Charleston, SC 29425, USA. [79]Nathan Kline Institute, Orangeburg, NY 10962, USA. [80]University of Iowa College of Medicine, Iowa City, IA 52242, USA. [81]University of South Florida: USF Health Byrd Alzheimer's Institute, Tampa, FL 33613, USA. [82]Department of Defense, Arlington, VA 22350, USA. [83]Stanford University, Stanford, CA 94305, USA.

