## [Peer Review File · Communications Medicine]

Reviewers' comments:

Reviewer #1 (Remarks to the Author):

This is an outstanding paper of major importance to those seeking to apply deep learning to polygenic risk prediction of Alzheimer's risk at the individual level. The approach is highly innovative and carefully considered. The data are clear, convincing, and somewhat straightforward. This last point is where I would like to see more nuanced reasoning or anticipation of what might be yet to come.

For example, Neff and colleagues (and others) have identified multiple subtypes of Alzheimer's disease. Surely, if these subtypes exist at the postmortem neuropathological level, then there must be heterogeneity in the predictors when the polygenic risk scores are estimated. How many subtypes are anticipated by the Ip analysis? Do the authors assume that the Chinese populations that they study are relatively homogeneous when it comes to gene pool and for that reason, the "Ip Analysis" truly does not reveal subtypes either in neuropathology or in polygenic risk scores?

Which neuropathological features are most determinative when it comes to the Ip Analysis? The best correlate for contemporaneous cognitive status at the time of death is synaptophysin content (synaptic density). Wouldn't the Ip Analysis be more valuable if synaptic density were one of the key outcomes?

These outstanding issues can be handled in the Discussion of this paper. No further experimentation is obviously required.

Reviewer #2 (Remarks to the Author):

Zhou et al. use deep learning methods in an attempt of improving polygenic risk analysis and prediction of Alzheimer's disease. The manuscript is complex and the rationale of modeling PRS using three non-independent datasets from a different ethnic origin is not very convincing. I consider that authors should modify the rationale trying to start defining a very robust PRS (functioning similarly in all tested datasets) in advance of launching secondary analyses for dissecting AD pathobiology.

Comments:

1. The introduction is pretty long and literature references are not updated.
2. Key and important references are emerging in Alzheimer's PRS field that should be cited. The results of this manuscript must be compared with contemporary PRS studies.
3. Please explain the reasoning for using the Jansen manuscript as the main reference for SNP rank selection. Also, reinforce the potential impact of using proxy-AD to select AD loci (Jansen paper used UKbiobank proxy cases). Of note, well-powered studies appearing afterward failed to confirm a relatively big proportion of signals reported by Jansen et al. Even the same research group has published a recent update (not cited in the text) with more than 1M individuals failing to replicate at least five loci reported in their previous release (ref. 23).

4. Please explain the reasoning for using $1E-08$ as a threshold instead of the conventional GWAS-significance threshold.
5. Illustrate the PRS model performance with and without APOE in western datasets
6. A major limitation of using western datasets is that selected cohorts are in fact part of discovery studies used for SNP selection (Jansen and Kunkle's papers used IGAP datasets as part of their respective GWAS). Hence, used datasets (ADNI, Nia, and ADC) are not fully independent of the reference ranking used for PRS modeling exploration. This important question should be discussed and appropriately addressed.
7. In the results sections, the authors mentioned that population structure co-variation was applied to Chinese datasets. However, it is not clear reading the paper results that the same covariates have been used in the Western datasets (ADNI, NIA, ADC) nor common demographics data (age, gender, or education). Reading the methods, it is clarified but I would recommend mentioning this point earlier in the text.
8. Please explain the reason why you use only 37 variants in Chinese datasets. To make comparisons, it would be better to apply identical methods to all the datasets explored. Furthermore, many true hits might not be significant (simply due to lack of power in Chinese datasets) but combined might increase PRS performance. I would suggest trying to reproduce identical PRS skipping the replication step and compare the results with 37-variants PRS.
9. In the case of Chinese studies, I would recommend the following experiments:
 - a. Use western rankings and different statistical models, evaluate their efficiency (or lack of efficiency) in WGS1 and WGS2 datasets (by applying identical modeling).
 - b. Alternatively, generate a Chinese ranking using WGS1 (but using an identical method used for analyzing Western datasets) and validate the results in WGS2.
 - c. Compared obtained ranking in the Chinese population with western rankings using genetic correlation approaches
 - d. Compare all models
 - i. With and without stratification covariates
 - ii. With and without demographics covariates (age, gender, education)
 - iii. With and without APOE
10. The authors also examined the association between individuals' scores and their cognitive functioning. However, they are using the same individuals for this study, and cognition performance is highly correlated with disease status (it can be considered a proxy or a surrogated marker). Hence it is predictable a similar performance of these dependent variables. So, the sentence: "Hence, the neural network risk scores calculated herein can predict cognitive function in the Chinese population" looks like a severe overstatement of this manuscript. The main reason is that these observations cannot be considered independent of case status analyses. The same is happening when analyzing WS2 endophenotypes that largely depend on case status (ie AD biomarkers). All observations described afterward can be considered highly predictable in my opinion.
11. Most probably, because of lack of power, the hits obtained only using WSG1 and WSG2 datasets might be false positive hits and this might be perturbing subsequent analyses as well.
12. Please Revise these sentences
"Importantly, as most genetic variants are present in the human genome...", I wonder if we might have genetic variants outside our genome!

“...because the prevalence of APOE-ε4 is only approximately 20%...” -> Please use a more appropriate lexicon (ie allele frequency, % allele carriers, etc).

Reviewer #3 (Remarks to the Author):

Zhou et al. tackle the problem of calculating polygenic risk scores (PRS) and case prediction for Alzheimer’s Disease (AD) patients. To this end, they apply a neural network model, examining data from several publicly available cohorts (ADNI, LOAD, ADC), as well as two Chinese cohorts. Unfortunately, this manuscript does not make a clear case for the advantages of their model against current state of the art PRS methods, the evaluations need to be more systematic, and the interpretability of their neural network model is also dubious.

Major comments:

1. In the intro (line 115), the authors posit that “the currently used PRS model assumes that the risk effects of the studied variants are independent.” This statement is untrue. There have been numerous proposed PRS methods, several of which that model the interdependency of variants, most notably taking into account LD structure. There are also other methods that incorporate functional priors as well (see some examples below).

The authors proceed to discuss a weighted PRS model and lasso-based model to be used as baseline methods in comparison to the neural network model they propose. While these methods are viable, to be fully convincing of the utility of their method, it would be critical to actually compare with other state of the art methods. A few notable examples include:

- * LDpred (Vilhjálmsson et al. 2015)
- * winner’s curse correction (Shi et al. 2016)
- * AnnoPred (Hu et al. 2017) * SBayesR (Lloyd-Jones et al. 2019)

2. Even within the methods that the authors set up as baselines, it seems as though the data used as input into the methods was different? They state that the weights for the weighted PRS model were based on summary statistics from Jansen et al, while the lasso and neural network models were trained via cross validation from 2 of the 3 AD cohorts.

3. The input SNPs also seemed to vary quite a bit from evaluation to evaluation and the seemingly ad hoc criteria used to select them is concerning, as it does not inspire confidence in the generalizability of their methods. For example, when moving to the Chinese populations, it appears as though the authors assembled a very specific set of 37 AD-associated SNPs chosen by various cutoffs across cohorts using logistic regression for the lasso and neural network models only, versus simply using thresholding of summary statistics as they did for the earlier evaluations.

4. One of the biggest concerns of neural network models is their “black box” nature and the lack of interpretability. The authors seem to flip this around by proposing that the modules in the penultimate layer represent biological processes. In their methods section, the authors even clarify that the number of nodes in this layer (5) is arbitrary corresponding to the potential number of pathways affecting the parallel. What happens when this number varies? What happens with more random input? The enrichment analyses are not convincing enough as they stand.

Minor comments:

1. Typo in line 196? Assuming the authors meant logistic regression and not logic regression?

Reviewer #4 (Remarks to the Author):

This paper benchmarked three models for AD risk prediction from population genotype data (SNPs): polygenic risk score (PRS), lasso model and neural network model. The authors first applied the models to public data from three cohorts, ADC, LOAD, ANDI, and then to their own private data from two Chinese cohorts. Both applications show that the neural network model outperforms two others for predicting AD. Using AD endophenotypic data from one of Chinese cohorts, they further found that the neural network based predictive scores highly correlate with many Ad biomarkers like proteins. Also, based on the neural network connectivity, they clustered input SNPs into several groups, each of which has been found to associate with cell-type related proteins. The manuscript was written clearly and organized logically. However, I have some major concerns, especially on neural network modeling.

- (1) The authors pointed out that PRS assumes that variants are independent, which I agree. However, the neural network model in this paper is simply feed-forward and just input individual snps, without modeling any relationships like epistatic effects. Many deep learning models such as graph neural network enable modeling such feature relationships. Also, it is not surprising that neural network beats linear weighted sums like PRS and LASSO for capturing higher order nonlinear groupings;
- (2) It is good to see that the authors use independent datasets to validate the trained models (ADNI to ADC+LOAD, 2nd Chinese cohort to 1st one). However, when they trained the model, they seem not use cross-validation. Instead, they just split 70% training and 30% validation data once. The one-time split highly likely has overfitting issues. The authors should use k-fold cross-validation. Also, many details on training neural networks are missing such as how to do early stopping, tune hyperparameters, etc;
- (3) The Chinese cohorts have whole genome sequencing data. It is very surprised that only 37 SNPs are found to associate with AD. Also, the Chinese samples are imbalanced (much more controls than ADs), so AUPRC is better to evaluate the performance than AUROC in such cases. The authors should report AUPRCs like they did for public data;
- (4) It is unclear how to link 37 snps to 5 hidden nodes at the 2nd last layer. What activation function(s) did the neural network use? Typically, people can prioritize input nodes for outputs by weights of neural network model such as via integrated gradients (Captum);
- (5) Also, the methods how to associate risk scores with cognitive measurements and endophenotypes are unclear. Did the authors consider any confounding factors since many endophenotypes contribute to AD;
- (6) The authors hypothesized gene regulatory mechanisms from those variants to gene expression at the cell type level. Can the authors check any regulatory variants such as if those variants are coding or if noncoding, do they interrupt TF binding sites on the regulatory elements to the genes? Many scRNA-seq and scATAC-seq data are available, enabling such mechanistic checking;
- (7) The authors should fix typos, inconsistent info (e.g., 5-layer vs. 7-layer), etc.

Following please find our point-by-point responses to the reviewers' comments:

Reviewer #1 (Remarks to the Author):

This is an outstanding paper of major importance to those seeking to apply deep learning to polygenic risk prediction of Alzheimer's risk at the individual level. The approach is highly innovative and carefully considered. The data are clear, convincing, and somewhat straightforward. This last point is where I would like to see more nuanced reasoning or anticipation of what might be yet to come.

For example, Neff and colleagues (and others) have identified multiple subtypes of Alzheimer's disease. Surely, if these subtypes exist at the postmortem neuropathological level, then there must be heterogeneity in the predictors when the polygenic risk scores are estimated. How many subtypes are anticipated by the Ip analysis? Do the authors assume that the Chinese populations that they study are relatively homogeneous when it comes to gene pool and for that reason, the "Ip Analysis" truly does not reveal subtypes either in neuropathology or in polygenic risk scores?

Which neuropathological features are most determinative when it comes to the Ip Analysis? The best correlate for contemporaneous cognitive status at the time of death is synaptophysin content (synaptic density). Wouldn't the Ip Analysis be more valuable if synaptic density were one of the key outcomes?

These outstanding issues can be handled in the Discussion of this paper. No further experimentation is obviously required.

We thank the reviewer for the kind suggestion. Our previous findings on plasma biomarkers associated with AD suggest that Alzheimer's disease (AD) polygenic risk affects multiple biological pathways and exerts heterogeneous effects. Regarding neuropathological features, we found that polygenic risk scores (PRSs) are significantly associated with white matter hyperintensity, which is a marker of demyelination and axonal loss in the brain ($p = 2.69E-02$; **Supplementary Figure 18; Supplementary Table 28; lines 354–357**).

To examine how the polygenic risk scores predict the heterogeneity in the predictors when they are estimated, we conducted additional analysis. Accordingly, in the revised manuscript, we performed unsupervised clustering methods using the outputs from the penultimate layer of the neural network model (**Figure 6; lines 430–439**). We showed that:

1. The individuals can be classified into 5 groups.
2. Compared with the normal controls (NCs) from those groups with lower AD risk, the NCs in high AD risk groups have:
 - Lower plasma amyloid-beta ($A\beta$) level ($p < 0.05$).
 - Increased plasma p-tau181 (tau phosphorylated at threonine-181) level.
 - Increased NfL (neurofilament light polypeptide) level.

3. The NCs from those groups with higher AD risk also exert altered expression in molecular pathways involving axon, neuron projection, and receptor activity.

These observations corroborate the reviewer's comments on the heterogeneity of the disease. Accordingly, in the *Discussion* section, we discuss the heterogeneity in disease polygenic risk and its possible contributions to the neuronal functions (**lines 472–479**).

Reviewer #2 (Remarks to the Author):

Zhou et al. use deep learning methods in an attempt of improving polygenic risk analysis and prediction of Alzheimer's disease. The manuscript is complex and the rationale of modeling PRS using three non-independent datasets from a different ethnic origin is not very convincing. I consider that authors should modify the rationale trying to start defining a very robust PRS (functioning similarly in all tested datasets) in advance of launching secondary analyses for dissecting AD pathobiology.

We thank the reviewer for the comments. As reported in the original manuscript, we conducted a polygenic score analysis on 3 AD cohorts recruited from European-descent populations. Based on the results from using weighted PRS models, we demonstrated that deep learning models classify AD risk more accurately than the weighted PRS models. We further showed similar observations by constructing PRS models in the 2 Chinese datasets.

To address the reviewer's suggestions and define a robust PRS analysis, we added the following new analyses in the revised manuscript:

1. We added weighted PRS results based on summary statistics from 2 separate AD genome-wide association studies (GWASs) from European-descent populations (Jansen et al.'s and Kunkle et al.'s; **Comment 3**) and 1 AD GWAS from the Chinese population (Zhou et al.'s; **Comment 9**).
2. We added new analyses on the performance of PRS models using variants outside of the APOE locus (chr19:44000000-46000000; GRCh38; **Comments 5, 9**) and PRS models using variants after linkage disequilibrium (LD)-based clumping.
3. We added new analyses using state-of-the-art PRS models including LDpred, Winner's curse correction, AnnoPred, and SBayesR methods.
4. We added new analyses comparing model performance in the 2 following conditions (**Comments 6, 7**):
 - a. Using all the samples from the 3 AD cohorts (ADC, LOAD, and ADNI) as training data.
 - b. Using all the samples from the 3 AD cohorts for 5-fold cross-validation analysis.
5. We evaluated the performance of PRS modeling in association with demographics (age, sex, and ethnic background; **Comment 9**).

6. We evaluated the performance of the models constructed using different sets of variants (listed below) in both European-descent cohorts and Chinese cohorts (**Comments 8, 9, and 11**), including:
 - The 1,799, 2,959, and 8,100 variants selected by p -value thresholds in Jansen et al.'s GWAS result
 - The 37 variants selected based on both Jansen et al.'s GWAS result and association results in Chinese AD cohorts
7. We re-examined the associations between the polygenic scores and cognitive performance in patients with mild cognitive impairment (**Comment 10**).

These newly conducted analyses confirmed our previous findings that deep learning models perform better in classifying AD risk than weighted PRS and lasso models. Furthermore, we showed that the 37 variants (we identified to be associated with AD in both European-descent cohorts and Chinese cohorts) can be used to classify AD risk in both European-descent and Chinese populations. Detailed responses to each comment are provided below.

Comments:

1. The introduction is pretty long and literature references are not updated.

We thank the reviewer for the comment. In the *Introduction* section of the revised manuscript, we removed some unnecessary information and updated the references for (1) recent research on AD PRS analysis (**lines 104–107**), and (2) recently developed weighted PRS models (**lines 114–119**).

2. Key and important references are emerging in Alzheimer's PRS field that should be cited. The results of this manuscript must be compared with contemporary PRS studies.

As suggested by the reviewer, we updated the revised manuscript to include the recent literature on AD PRS studies (**lines 104–107**). Meanwhile, we revised the corresponding sentences in the *Discussion* section and discussed the comparisons between the results from our deep learning model and the contemporary PRS studies as follows:

“In classifying the clinically diagnosed AD patients, the best auROC our neural network model achieved was 0.84, which is higher than other recently reported results based on the weighted PRS model (auROC = 0.74; PMID: 34301930).” (**lines 466–468**)

3. Please explain the reasoning for using the Jansen manuscript as the main reference for SNP rank selection. Also, reinforce the potential impact of using proxy-AD to select AD loci (Jansen paper used UKbiobank proxy cases). Of note, well-powered studies appearing afterward failed to confirm a relatively big proportion of signals reported by Jansen et al. Even the same research group has published a recent update (not cited in the text) with more than 1M individuals failing to replicate at least five loci reported in their previous release (ref. 23).

We thank the reviewer for the kind suggestion. We selected Jansen et al.'s GWAS results for constructing the PRS model, despite the study having proxy cases, because of its greater sample size compared with other AD GWASs published in a similar period. This should provide a more unbiased estimation of the contribution of individual variants' risk effects on AD. Meanwhile, as genetics contribute to 60%–80% of AD risk, it is likely that proxy cases share a high similarity with the clinically confirmed cases. Moreover, to select the variants, we used geometric sequences (1E-4, 1E-6, and 1E-8) of the p -values (1E-4, 1E-6, and 1E-8) instead of a genome-wide significant threshold; thus, it is likely that most of the variants that contribute to AD were included in the model construction when p -values 1E-4 were used.

In the revised manuscript, we also constructed the weighted PRS model using the GWAS summary statistics from another study, by Kunkle et al. (PMID: 30820047), which has a smaller number of AD patients, although all were clinically confirmed AD cases. By comparing the results obtained from the models constructed based on Jansen et al.'s and Kunkle et al.'s data, respectively, we showed that the weighted PRS models from these 2 GWAS datasets do not exert a significant difference in their accuracy of classifying AD risk (**Supplementary Figure 1; Supplementary Table 3; lines 164–169**). Hence, we believe the use of Jansen et al.'s GWAS data to construct the weighted PRS model is appropriate for this study.

4. Please explain the reasoning for using 1E-08 as a threshold instead of the conventional GWAS-significance threshold.

We thank the reviewer for the comment. The p -value threshold of 1E-8 is frequently used in the existing literature when investigating the performance of PRS models (e.g., PMID: 26430803, 34628483, and 30992449). In this study, we used a geometric sequence (1E-4, 1E-6, and 1E-8) of the p -values to obtain a different number of variants for the PRS model construction.

5. Illustrate the PRS model performance with and without APOE in western datasets

As suggested by the reviewer, in the revised manuscript, we added a new analysis of the performance of PRS models using variants outside of the *APOE* locus (chr19:44000000-46000000; GRCh38).

In brief, our updated data suggest that the variants outside of the *APOE* locus still significantly contribute to AD risk, as shown by $\text{auROC} = 0.57\text{--}0.59$ in the obtained weighted PRS models (**Supplementary Figure 1; Supplementary Table 3; lines 161–163**). Moreover, our 5-fold cross-validation analysis results also suggest PRS models constructed based on variants outside of the *APOE* locus can classify AD, including weighted PRS ($\text{auROC} = 0.57\text{--}0.60$), lasso ($\text{auROC} = 0.58\text{--}0.60$), and neural network models ($\text{auROC} = 0.58\text{--}0.59$) (**Supplementary Table 6**). Notably, these results are comparable to the observations made in the contemporary PRS study reported by Leonenko et al. ($\text{auROC} = 0.56\text{--}0.61$; PMID: 34301930).

6. A major limitation of using western datasets is that selected cohorts are in fact part of discovery studies used for SNP selection (Jansen and Kunkle's papers used IGAP datasets as part of their respective GWAS). Hence, used datasets (ADNI, Nia, and ADC) are not fully independent of the reference ranking used for PRS modeling exploration. This important question should be discussed and appropriately addressed.

We thank the reviewer for the kind comments. The reviewer suggested that part of the samples used to assess the model performance may have also been used in the previous GWASs, where the effect size was obtained for the construction of weighted PRS models. Of note, this would only cause the obtained weighted PRS model to display a higher classification accuracy (owing to possible overfitting), while it would not affect the performance of other models (i.e., the lasso and neural network models). Given that our major findings show the superior performance of the neural network model in classifying disease risk over the weighted PRS model, this limitation should not affect our major conclusions.

Meanwhile, in the revised manuscript, we added a new analysis evaluating the model performance by constructing models based on study samples (effect sizes were estimated by the logistic regression in the same samples). These results show that using the neural network model leads to higher AD classification accuracy (**Supplementary Figure 2; Supplementary Table 5; lines 188–199**). In addition, the results from the 5-fold cross-validation analysis we conducted using beta from our analyzed dataset also support the superiority of the neural network model in classifying disease risk compared with the weighted PRS model (**Supplementary Figure 3; Supplementary Table 6; lines 200–207**).

7. In the results sections, the authors mentioned that population structure co-variation was applied to Chinese datasets. However, it is not clear reading the paper results that the same covariates have been used in the Western datasets (ADNI, NIA, ADC) nor common demographics data (age, gender, or education). Reading the methods, it is clarified but I would recommend mentioning this point earlier in the text.

We thank the reviewer for pointing this out. In the previous version of our manuscript, we conducted logistic regression to evaluate whether AD risk variants reported in the European AD GWASs exert an AD association in the Chinese datasets, using the covariates age, gender, and population structure. We also applied the same covariates in the logistic regression to regress out their confounding effects before conducting the lasso modeling in both the Chinese and European-descent datasets.

In the revised manuscript, we added new analyses in the European-descent datasets and included the same covariates (**lines 188–207**). Meanwhile, we revised the *Methods* section and stated the usage of those covariates in all the analyses accordingly.

8. Please explain the reason why you use only 37 variants in Chinese datasets. To make comparisons, it would be better to apply identical methods to all the datasets explored. Furthermore, many true hits might not be significant (simply due to lack of power in Chinese datasets) but combined might increase PRS performance. I would suggest trying to reproduce identical PRS skipping the replication step and compare the results with 37-variants PRS.

We thank the reviewer for the kind comments and suggestions. We selected the 37 variants based on their significant associations with AD in the Chinese population, which provides a higher chance to capture AD risk in the Chinese population.

In the revised manuscript, we applied the same models constructed from the European-descent population (variants selected by p -value thresholds) to the Chinese datasets. We also used the same sets of variants to construct PRS models in the Chinese datasets. Furthermore, we compared the model performance obtained from variants selected by p -value thresholds and the 37 variants. Our current data suggest that:

1. For models constructed in the European-descent population using 1,799, 2,959, and 8,100 variants, the weighted PRS model showed poor classification accuracy in both Chinese WGS cohorts 1 and 2 (auROCs = 0.50), while the lasso and neural network models classified disease risk to a certain degree (auROCs > 0.63) (**Supplementary Figure 9; Supplementary Tables 12–15; lines 250–252**).
2. In the European-descent data (**Supplementary Figure 13**):
 - a. The neural network model showed superior performance in classifying patients with AD as compared to the weighted PRS models when 37, 1,799, 2,959, or 8,100 variants were used.
 - b. For any of the 3 types of models, the model constructed from 37 variants exhibited a comparable performance as compared to the models of the same types constructed from 1,799, 2,959, and 8,100 variants.
 - c. Particularly, the weighted PRS and lasso models constructed with 37 variants had significantly higher AD classification accuracy than the models of the same types constructed with 1,799, and 2,959 variants.
3. In the Chinese data (**Supplementary Figure 14**):
 - a. The neural network models constructed from 37 variants exhibited comparable (or higher) performance as compared with the models of the same types constructed from 1,799, 2,959, and 8,100 variants (**lines 302–306**).
4. When comparing the models based on Chinese and European-descent populations using 1,799, 2,959, and 8,100 variants, in both WGS1 and WGS2 cohorts, the Chinese weighted PRS models were more accurate than the European-descent weighted PRS models. Meanwhile, in the WGS1 cohorts, the lasso and neural network models constructed based on WGS1 data more accurately predicted AD than models based on European-descent data (**Supplementary Figure 9; Supplementary Tables 12–15**).

As the current manuscript focuses on the utility of the neural network model in disease risk prediction, we presented all these results in the *Supplementary Materials* (**Supplementary Figures 9, 13, 14; Supplementary Tables 12–15**) and discussed the most relevant results in the main text accordingly (**lines 250–252; 302–306**).

9. In the case of Chinese studies, I would recommend the following experiments:

- a. Use western rankings and different statistical models, evaluate their efficiency (or lack of efficiency) in WGS1 and WGS2 datasets (by applying identical modeling).
- b. Alternatively, generate a Chinese ranking using WGS1 (but using an identical method used for analyzing Western datasets) and validate the results in WGS2.
- c. Compared obtained ranking in the Chinese population with western rankings using genetic correlation approaches
- d. Compare all models
 - i. With and without stratification covariates
 - ii. With and without demographics covariates (age, gender, education)
 - iii. With and without APOE

We thank the reviewer for the kind inspection and suggestion. We conducted all the suggested analyses in the revised manuscript and found that:

1. For the weighted PRS, lasso, and neural network models, the model performance was similar between males and females (**Supplementary Figure 7; Supplementary Table 10**), while varied performance may be observed among the different ethnic backgrounds (**Supplementary Figure 6; Supplementary Tables 9**) or age groups (**Supplementary Figure 8; Supplementary Table 11**). Due to the limited sample size in the Chinese population, we only examined the effects of these confounding factors in the European-descent population in the revised manuscript (**lines 227–243**).
2. For our transethnic analysis, we showed that:
 - a. For models constructed in the European-descent population using 1,799, 2,959, and 8,100 variants, the weighted PRS model showed poor classification accuracy in both Chinese WGS cohorts 1 and 2 (auROCs = 0.50), while the lasso and neural network models could classify disease risk to a certain degree (auROCs > 0.63) (**Supplementary Figures 9, 10; Supplementary Tables 12–15; lines 250–252**).
 - b. In the WGS1 data, among PRS models based on 1,799 and 2,959 variants, the neural network model exerted superior performance in disease risk classification as compared with the weighted PRS model (**Supplementary Figure 14**).
 - c. For models constructed using 1,799, 2,959, and 8,100 variants, in both WGS1 and WGS2 cohorts, the weighted PRS models constructed based on Chinese data (effect sizes from WGS1 and Chinese AD GWASs) showed superior performance in classifying AD when compared with the models based on European-descent data. Meanwhile, in the WGS1 cohort, the lasso and neural network models constructed based on WGS1 data showed superior performance when compared with the models of the same types based on European-descent data (**Supplementary Figure 9; Supplementary Tables 12–15**).

- d. Genomic correlation analysis suggests that only the lasso and neural network models obtained from the 2 ethnic backgrounds capture similar disease variations ($R^2 > 0.50$) in the WGS1 data, while the polygenic scores between the weighted PRS models from the 2 ethnic backgrounds displayed little correlations in the WGS1 data ($R^2 < 0.01$) (**Supplementary Figure 10**).

We included all these results in the *Supplementary Materials* (**Supplementary Figures 6–10; Supplementary Tables 9–15**) in the revised manuscript. Meanwhile, we discussed the most relevant findings in the main text accordingly (**lines 227–243, 250–252**)

10. The authors also examined the association between individuals' scores and their cognitive functioning. However, they are using the same individuals for this study, and cognition performance is highly correlated with disease status (it can be considered a proxy or a surrogated marker). Hence it is predictable a similar performance of these dependent variables. So, the sentence: "Hence, the neural network risk scores calculated herein can predict cognitive function in the Chinese population" looks like a severe overstatement of this manuscript. The main reason is that these observations cannot be considered independent of case status analyses. The same is happening when analyzing WS2 endophenotypes that largely depend on case status (ie AD biomarkers). All observations described afterward can be considered highly predictable in my opinion.

We thank the reviewer for pointing this out. We only used the patients with AD and NCs in the WGS1 cohort to construct the PRS models. However, when studying the correlation between the polygenic scores and cognitive performance, we used the cognitive performance data from patients with AD, patients with mild cognitive impairment (MCI), and NCs in the WGS1 cohort (**Figures 3e–h; lines 333–338**) as well as data from the WGS2 cohort (**Supplementary Figure 16; lines 338–341**). Hence, the individuals used to construct the models and those used to assess the association between polygenic scores and cognitive performance are not the same.

To further strengthen our analysis on the association between individuals' scores and their cognitive functioning, in the revised manuscript, we included additional MCI patients from the WGS2 cohort in the analysis and provided data for the association results in MCI patients from the WGS1 cohort. Our new data again suggest the significant association between individuals' scores and their cognitive functioning in those MCI patients, implying the utility of polygenic scores in indicating individual cognitive functioning (**Supplementary Figure 17; lines 336, 341**).

11. Most probably, because of lack of power, the hits obtained only using WSG1 and WSG2 datasets might be false positive hits and this might be perturbing subsequent analyses as well.

We thank the reviewer for the comments. In the Chinese population, we chose the 37 variants used to establish the PRS models from the 216 variants in existing published AD GWASs, which also exhibited significant AD associations in the Chinese population. Therefore, we did not select the 37 hits solely based on the WGS1 and WGS2 datasets, and they should have a

higher chance to be AD-modifying genetic factors in both European-descent and Chinese populations (**lines 259–271**).

Of note, in the revised manuscript we showed that:

- 5-fold cross-validation analysis suggests that the PRS models based on these 37 variants can classify AD risk in both European-descent (auROC = 0.68–0.72) and Chinese (auROC = 0.66–0.69) populations (**Supplementary Figure 12; Supplementary Table 23**).
- The lasso and neural network models constructed from the 37 variants exhibited a comparable (or better) performance than the models constructed based on the variants selected by p -value thresholds (8,100, 2,959, and 1,799 variants) in both the European-descent and Chinese populations (**Supplementary Figures 13, 14**).
- The 37-variant PRS models constructed based on the European-descent data can classify AD risk in the Chinese population (auROC = 0.60–0.67, auPRC = 0.69–0.72), and *vice versa* (auROC = 0.62–0.65; auPRC = 0.70–0.73; **Supplementary Figure 15; Supplementary Table 24**).

Thus, the PRS models based on the 37 variants associated with AD in European-descent and Chinese populations can be used for modeling and classifying AD risk in both populations. We included these results in the main text accordingly (**lines 297–311**).

12. Please Revise these sentences

“Importantly, as most genetic variants are present in the human genome...”, I wonder if we might have genetic variants outside our genome!

“...because the prevalence of APOE-ε4 is only approximately 20%...” -> Please use a more appropriate lexicon (ie allele frequency, % allele carriers, etc).

We thank the reviewer for the kind comments. We revised the sentences accordingly as follows:

1) “Importantly, as most genetic variants are present in the human genome...” (**line 78**)

Revised to: “Importantly, as genetic variants are present in the human genome...” (deleted “most”).

2) “...because the prevalence of APOE-ε4 is only approximately 20%...” (**line 94**)

Revised to: “...the allele frequency of APOE-ε4 is only approximately 20% in populations of European descent and 10% in Asian populations...” (changed “prevalence” to “allele frequency”).

Reviewer #3 (Remarks to the Author):

Zhou et al. tackle the problem of calculating polygenic risk scores (PRS) and case prediction for Alzheimer's Disease (AD) patients. To this end, they apply a neural network model, examining data from several publicly available cohorts (ADNI, LOAD, ADC), as well as two Chinese cohorts. Unfortunately, this manuscript does not make a clear case for the advantages of their model against current state of the art PRS methods, the evaluations need to be more systematic, and the interpretability of their neural network model is also dubious.

Major comments:

1. In the intro (line 115), the authors posit that “the currently used PRS model assumes that the risk effects of the studied variants are independent.” This statement is untrue. There have been numerous proposed PRS methods, several of which that model the interdependency of variants, most notably taking into account LD structure. There are also other methods that incorporate functional priors as well (see some examples below). The authors proceed to discuss a weighted PRS model and lasso-based model to be used as baseline methods in comparison to the neural network model they propose. While these methods are viable, to be fully convincing of the utility of their method, it would be critical to actually compare with other state of the art methods. A few notable examples include:

- * LDpred (Vilhjálmsón et al. 2015)
- * winner's curse correction (Shi et al. 2016)
- * AnnoPred (Hu et al. 2017) * SBayesR (Lloyd-Jones et al. 2019)

We thank the reviewer for the suggestions. In the revised manuscript, we added an introduction for those state-of-the-art methods that can deal with LD information (**lines 112–116**). Meanwhile, we constructed PRS models using LDpred, Winner's curse correction, AnnoPred, and SBayesR methods and compared their performance with weighted PRS models. In brief, our analysis suggests that if using the variants selected by current p -value thresholds, then those state-of-the-art methods do not exert better performance in AD risk classification in the European-descent population as compared with the weighted PRS models we used in the previous version of our manuscript (**Supplementary Figure 1; Supplementary Table 4; lines 169–172**).

2. Even within the methods that the authors set up as baselines, it seems as though the data used as input into the methods was different? They state that the weights for the weighted PRS model were based on summary statistics from Jansen et al, while the lasso and neural network models were trained via cross validation from 2 of the 3 AD cohorts.

We thank the reviewer for the kind suggestions. In the previous version of the manuscript, we used different data as input to construct the models. For the weighted PRS models, it is a common strategy to use the effect sizes to select AD-associated variants to construct the weighted PRS model. For lasso and neural network models, we trained the model using raw

genotype data obtained from AD study cohorts, and determined the contribution of each variant to disease risk in the model training process.

To address the reviewer's concern and provide a fairer comparison, in the revised manuscript, we conducted new analyses by comparing the model performance using the data from the same groups of individuals. Specifically, we conducted the following two new analysis:

- (i) Using all the samples from the 3 AD cohorts (ADC, LOAD, and ADNI) as training data.
- (ii) Using all the samples from the 3 AD cohorts for 5-fold cross-validation analysis.

In our new analyses, the results again corroborated our previous findings, namely that the neural network model performs better than the lasso and weighted PRS models in estimating AD risk. ($p < 0.001$; **Supplementary Figures 2, 3; Supplementary Table 5, 6; lines 188–207**).

3. The input SNPs also seemed to vary quite a bit from evaluation to evaluation and the seemingly ad hoc criteria used to select them is concerning, as it does not inspire confidence in the generalizability of their methods. For example, when moving to the Chinese populations, it appears as though the authors assembled a very specific set of 37 AD-associated SNPs chosen by various cutoffs across cohorts using logistic regression for the lasso and neural network models only, versus simply using thresholding of summary statistics as they did for the earlier evaluations.

We thank the reviewer for the critical comments. The reason we selected those 37 variants is that we aimed to use variants associated with AD in the Chinese population to evaluate the performance of neural network models in the Chinese population.

In the revised manuscript, we conducted further analyses to improve the logical flow of the manuscript. We first showed that the weighted PRS model constructed in the cohorts exerts varied performance across different ethnic backgrounds (**Supplementary Figure 6; Supplementary Tables 9; lines 229–237**). Specifically, we showed that the weighted PRS model based on Jansen et al.'s AD GWAS summary statistics shows poor classification accuracy in the Chinese population (both WGS1 and WGS2 cohorts; auROCs ~ 0.50 ; **Supplementary Figures 9, 10; Supplementary Tables 12–15; lines 250–252**). These results suggest that the variants selected from AD GWASs in European-descent populations may not be suitable for modeling AD risk in Chinese. Hence, we conducted an analysis to identify variants that exert an AD association in Chinese and identified 37 variants for modeling disease risk.

Of note, in the revised manuscript, we showed that:

1. Our 5-fold cross-validation analysis suggests that the PRS models based on these 37 variants can classify AD risk in both European-descent (auROC = 0.68–0.72) and Chinese (auROC = 0.66–0.69) populations (**Supplementary Figure 12; Supplementary Table 23; lines 299–302**).
2. The neural network model constructed using 37 variants distinguished patients with AD from NCs more accurately than the weighted PRS and lasso models in both European-

- descent and Chinese populations ($p < 0.001$; **Supplementary Figure 12; Supplementary Table 23**).
3. The neural network models constructed from 37 variants exhibited comparable (or better) performance than the models constructed from variants selected by p -value thresholds (8,100, 2,959, and 1,799 variants) in both the European-descent and Chinese AD cohorts (**Supplementary Figures 13, 14; lines 302–306**).
 4. The 37-variant PRS models constructed based on the European-descent data can classify AD risk in the Chinese population (auROC = 0.60–0.67, auPRC = 0.69–0.72), and vice versa (auROC = 0.62–0.65; auPRC = 0.70–0.73; **Supplementary Figure 15; Supplementary Table 24; lines 306–309**)

Hence, those data from the 37 variants further support our major conclusion, which is the utility of the neural network model in modeling disease polygenic risk. Meanwhile, the PRS models based on these 37 variants can be used for modeling and classifying AD risk in both populations.

4. One of the biggest concerns of neural network models is their “black box” nature and the lack of interpretability. The authors seem to flip this around by proposing that the modules in the penultimate layer represent biological processes. In their methods section, the authors even clarify that the number of nodes in this layer (5) is arbitrary corresponding to the potential number of pathways affecting the parallel. What happens when this number varies? What happens with more random input? The enrichment analyses are not convincing enough as they stand.

We thank the reviewer for the critical comments and suggestions. In the current analysis, we performed the association analysis between the polygenic scores and the plasma proteome data to assess the possible effects of AD polygenic risk on the biological processes. Specifically, our results suggest that AD polygenic risk has an effect on TNF- α - and cytokine-related pathways, which have also been reported in large, up-to-date AD GWASs (PMID: 35379992).

To strengthen our analysis, in the revised manuscript, we performed an additional analysis in which we changed the number of nodes in the penultimate layer ($n = 2, 3, \text{ and } 10$) and performed the association analysis between the outputs from the nodes in the penultimate layer and the plasma biomarkers (**Supplementary Figure 19; lines 412–424**).

First, we found that the neural network risk scores obtained from the modified models are highly correlated ($R^2 > 0.88$; **Supplementary Figures 19a and b**). In addition, these modified models can recover >80% of the plasma proteins previously identified as associated with the neural network risk scores (with p -values < 0.10 ; **Supplementary Figures 19c and d**). Furthermore, for the neural network model with 3 nodes in the penultimate layer, the multivariate analysis associating the plasma protein profiles with the outputs from the penultimate layer again highlighted the associations between polygenic risks and immune-associated signaling pathways, including TNF- α - and cytokine-related pathways (**Supplementary Figures 19e and f**). Therefore, these new findings further strengthen our conclusions that AD polygenic risk is associated with immune-associated pathways.

Moreover, in the revised manuscript, we provided additional data on the model interpretability, i.e., which variants play more vital roles in classifying AD in the neural network models, and

what might be the possible mechanism. Specifically, we show that coding variant rs429358, which encodes *APOE-ε4*, one of the most well-accepted AD genetic risks, is significantly correlated with the obtained score (Spearman's $\rho = 0.24$, $p < 0.001$; **Supplementary Figure 20a**). Meanwhile, rs439401, another noncoding variant and an AD risk factor that exerts a risk effect independent of the *APOE-ε4* genotype, is also significantly correlated with the score. Hence, our new data suggest that variants exerting specific biological functions may have greater effects on modulating disease risks, which is also useful in modeling disease risks (**Supplementary Figure 20; lines 440–461**).

Minor comments:

1. Typo in line 196? Assuming the authors meant logistic regression and not logic regression?

We thank the reviewer for the comment. We revised the text by changing the word “logic” to “logistic” (**line 262**).

Reviewer #4 (Remarks to the Author):

This paper benchmarked three models for AD risk prediction from population genotype data (SNPs): polygenic risk score (PRS), lasso model and neural network model. The authors first applied the models to public data from three cohorts, ADC, LOAD, ANDI, and then to their own private data from two Chinese cohorts. Both applications show that the neural network model outperforms two others for predicting AD. Using AD endophenotypic data from one of Chinese cohorts, they further found that the neural network based predictive scores highly correlate with many Ad biomarkers like proteins. Also, based on the neural network connectivity, they clustered input SNPs into several groups, each of which has been found to associate with cell-type related proteins. The manuscript was written clearly and organized logically. However, I have some major concerns, especially on neural network modeling.

(1) The authors pointed out that PRS assumes that variants are independent, which I agree. However, the neural network model in this paper is simply feed-forward and just input individual snps, without modeling any relationships like epistatic effects. Many deep learning models such as graph neural network enable modeling such feature relationships. Also, it is not surprising that neural network beats linear weighted sums like PRS and LASSO for capturing higher order nonlinear groupings;

We thank the reviewer for the critical comments. We agree with the reviewer that the neural network, with its sophisticated, multi-layered architecture, is more suitable for capturing higher order nonlinear groupings than the PRS and lasso models. Particularly, in the current study, we modeled the disease risk using hundreds or thousands of variants. Of note, the epistasis suggests that the individual variants' effects on the disease are not fixed and may be affected by the presence of other variants. Thus, different combinations of variants may have different contributions to disease risk. Such variations cannot be adequately modeled by the weighted PRS and lasso models, while they can likely be modeled by the multilayer perceptron (MLP) model, as the fully connected layers in the MLP model should be able to model all possible

variant combinations, capturing key combinations and estimating their effects on AD risk during the training.

However, we acknowledge that the MLP model cannot directly model the interactions between the variants. Thus, in the revised manuscript, we also evaluated the possible utility of the graph neural network (GNN) model in classifying disease risk (**Supplementary Figure 21**).

In brief, we regarded the disease classification problem as a graph classification problem: each participant is represented as a graph, where nodes refer to the selected variants, edges refer to the LD between the variants, and graph labels refer to the phenotypes (**Supplementary Figure 21a**). Notably, by conducting 5-fold cross-validation analysis for the GNN model based on the selected 37 variants, we achieved an AD classification accuracy at an auROC of 0.71, which is better than that of the weighted PRS model (auROC = 0.68; $p < 0.001$; **Supplementary Figure 21b**). Hence, our results further demonstrate the utility of the GNN model in modeling disease risk. The details can be found in the *Methods* (**lines 621–640**) and *Discussion* sections accordingly (**lines 493–503**).

(2) It is good to see that the authors use independent datasets to validate the trained models (ADNI to ADC+LOAD, 2nd Chinese cohort to 1st one). However, when they trained the model, they seem not use cross-validation. Instead, they just split 70% training and 30% validation data once. The one-time split highly likely has overfitting issues. The authors should use k-fold cross-validation. Also, many details on training neural networks are missing such as how to do early stopping, tune hyperparameters, etc;

We thank the reviewer for the critical suggestions. To address the reviewer's concern and provide a fairer comparison, in the revised manuscript, we adopted a 5-fold cross-validation analysis to compare the performances of the weighted PRS, lasso, and neural network models using the same grouping strategy for separating the training and testing samples.

The results of the 5-fold cross-validation analysis corroborate our previous observations, i.e., the neural network model exerts a superior performance in classifying AD risk as compared with both the lasso and weighted PRS models in the European-descent AD cohorts ($p < 0.001$; **Supplementary Figures 3, 11; Supplementary Table 6; lines 200–207**). Meanwhile, when we used 37 variants to construct the models, the 5-fold cross-validation analysis again suggested that the neural network model exerts a superior performance in classifying AD risk as compared with both the lasso and weighted PRS models in the European-descent and Chinese AD cohorts (**Supplementary Figure 12; Supplementary Table 23**).

In addition, we revised the *Methods* section by adding further details on the parameters used to construct the neural network models (**lines 588–620**).

(3) The Chinese cohorts have whole genome sequencing data. It is very surprised that only 37 SNPs are found to associate with AD. Also, the Chinese samples are imbalanced (much more

controls than ADs), so AUPRC is better to evaluate the performance than AUROC in such cases. The authors should report AUPRCs like they did for public data;

We thank the reviewer for the kind suggestion. To select the 37 variants, we first identified 216 AD GWAS hits published in large scale GWASs (PMID: 35379992, 33589840, 34493870, 30617256, 30820047, 29777097, 33074286, 33188687, and 28183528), and later found that only 37 variants were associated with AD in the Chinese population. Thus, we suggested that these variants are important for modulating AD risk and will be suitable for modeling the disease polygenic risk. The reason why only 37 variants from 216 were replicated in Chinese may be due to: 1) AD risk variants identified in European-descent populations do not exert similar disease risk in other populations (e.g., *APOE-ε4* has a lower AD risk in African-descent populations when compared with that in European-descent populations; PMID: 30517106); 2) Disease risk of the same AD risk locus may be contributed by distinct variants in different ethnic groups (e.g., *TREM2* R47H rare coding variant is more prevalent in European-descent populations, while H157Y is more frequently found in Asian populations; PMID: 32775599)

Moreover, in the revised manuscript, we followed the reviewer's suggestion and added the auPRC results for the Chinese dataset (**Figure 3b**). Meanwhile, we provided results on the auPRCs for the newly added analysis (please see the updated data in the *Supplementary Materials*).

(4) It is unclear how to link 37 snps to 5 hidden nodes at the 2nd last layer. What activation function(s) did the neural network use? Typically, people can prioritize input nodes for outputs by weights of neural network model such as via integrated gradients (Captum);

We thank the reviewer for the critical comments. We used the sigmoid function as the activation function for the neural network models in this study (**line 600**). In addition, in the *Methods* section of the revised manuscript, we provided more details on the methodology for model construction.

As for the model interpretation, in the revised manuscript, we conducted a partial correlation analysis between the genotype dosage of input variants and the final polygenic score. This analysis identified variants that play more weight in contributing to the final risk score (**Supplementary Figure 20**). Meanwhile, by annotating variants with their putative biological functions, we showed that those identified variants also exert certain biological functions (residing in coding regions that may alter protein function, or in regulatory regions that may modulate gene expression; **lines 441–461**). We have further discussed the potential utility of model interpretability in disease mechanism studies in the *Discussion* section accordingly (**lines 493–503**).

(5) Also, the methods how to associate risk scores with cognitive measurements and endophenotypes are unclear. Did the authors consider any confounding factors since many endophenotypes contribute to AD;

Yes, when performing the association analysis between polygenic scores and cognitive performance as well as plasma biomarkers, we considered covariates including age, gender, and genomic structure (represented by the top 5 principal components in the genotype data). We included this information in the corresponding *Methods* section of the revised manuscript (lines 690–694).

(6) The authors hypothesized gene regulatory mechanisms from those variants to gene expression at the cell type level. Can the authors check any regulatory variants such as if those variants are coding or if noncoding, do they interrupt TF binding sites on the regulatory elements to the genes? Many scRNA-seq and scATAC-seq data are available, enabling such mechanistic checking;

We thank the reviewer for the kind suggestion. In the revised manuscript, we annotated the biological functions of the selected 37 variants, including their possible localization in the coding, untranslated, open chromatin (enriched for DNA regulatory elements), polymerase binding, and transcription factor binding regions. (Supplementary Figure 20a; lines 441–446).

Our data suggest that variants with more contributions to the final risk score are associated with specific biological functions. For instance, the noncoding variant rs439401, previously identified as an AD risk factor that exerts a risk effect independent of the *APOE-ε4* genotype (PMID: 31346172), is significantly correlated with the score (Spearman's $\rho = 0.05$, $p < 0.001$; Supplementary Figure 20a). Of note, rs439401 resides in the regulatory region and occupies the transcription factor binding regions, which may influence the gene expression (Supplementary Figure 20b, c). Furthermore, our genotype-expression association analysis revealed the association between rs439401 and the altered APOE expression in skin tissues (Supplementary Figure 20d). Meanwhile, brain single-cell ATAC-seq (Assay for Transposase-Accessible Chromatin using sequencing) data suggest that variant rs439401 resides in the open chromatin regions, suggesting the roles of rs439401 in modulating APOE expression in the brain (Supplementary Figure 20e; data from PMID: 33106633). More detailed results can be found in the subsection “*Modeling of disease risk by polygenic score*” of the revised manuscript (lines 440–461).

(7) The authors should fix typos, inconsistent info (e.g., 5-layer vs. 7-layer), etc.

We thank the reviewer for the careful review. It should be “7-layer,” and we have fixed the typos.

Reviewers' comments:

Reviewer #1 (Remarks to the Author):

The authors have provided responses and revisions that, in my opinion, more than adequately address the concerns expressed by all the referees in the initial round of review. Certainly they have addressed my concerns.

Given the completeness of the responses, I would support a decision to advance the paper forward in the process.

I have no new concerns to express.

Reviewer #4 (Remarks to the Author):

The authors well addressed most of my previous comments. I appreciated their graph neural network analysis, but using $LD R^2 > 0.6$ to link variants sounds arbitrary and doesn't represent epistatic interactions, which may cause the weak performance of GNN in Fig S21. Also, both AUROC & AUPRC don't show significant outperformance of neural network models (e.g., 0.77 vs. 0.74). Please comment.

Also, if possible, run and compare the neural network models using 216 GWAS SNPs or randomly select 37 SNPs to see if those 37 variants are really Chinese-specific.

Here are my comments about how they addressed Rev 3.

1. Benchmarking state-of-the-art PRS methods. It is great to see they compared their weighted PRS model with several state-of-the-arts, which do not show any significant improvement over weighted PRS for classifying AD in European cohorts. However, is it possible to show the comparison on Chinese cohorts, which are the main selling point of this work?

2. Addressed.

3. The authors showed comparable performances between p-val select SNPs and "37 SNPs". This might be acceptable since 37 SNPs are much less dimensional (more represented features) than thousands of SNPs. Does the latter overlap or include the former? It is unclear if the authors have also fully optimized the neural network for p-val select SNPs (e.g., Fig S14) for a fair comparison. Actually, it is very surprise to see the neural network model with much less features (N=37) is significantly better than the ones with much more features. Moreover, how did the authors generate 37 variants from European cohort (Fig. S15)? Are they the same for Chinese cohorts?

4. Addressed.

5. Minor comment is fixed.

Reviewer #5 (Remarks to the Author):

The authors have gone to great lengths to address all the queries and recommendations put forward after the initial review.

In my opinion, these have been addressed to a satisfactory extent that justifies the publication of the revised version of the manuscript.

Please find our point-by-point responses to the reviewer's comments below:

Reviewer #4 (Remarks to the Author):

The authors well addressed most of my previous comments. I appreciated their graph neural network analysis, but using LD $R^2 > 0.6$ to link variants sounds arbitrary and doesn't represent epistatic interactions, which may cause the weak performance of GNN in Fig S21. Also, both AUROC & AUPRC don't show significant outperformance of neural network models (e.g., 0.77 vs. 0.74). Please comment.

We thank the reviewer for the detailed review and valuable comments.

In the current analysis, we chose LD $R^2 > 0.6$ because $R^2 \geq 0.6$ is commonly used as the criterion to determine whether SNP pairs are in (strong) linkage disequilibrium (PMID: 29459708, 31591465). We agree with the reviewer that the current GNN model does not outperform the neural network model in classifying AD risk. However, the purpose of conducting GNN analyses in the current study is to demonstrate that, besides multilayer perceptron, other deep learning models (e.g., GNN, RNN, CNN) can also be used to classify AD risk based on genomic variants.

In our revision, using an 80/20 split strategy with early stopping, we performed an additional analysis on the GNN models built based on the 37 selected genetic variants with different edge weights derived from the R^2 and l -norm distance between the feature vectors of the two endpoints of the underlying edge (**Table 1**). Our results demonstrated that those GNN models constructed based on the 37 variants showed similar performance in classifying AD risk in the European-descent population (auROC = 0.66–0.67), suggesting that there is limitation for further optimization of the GNN model's performance in using LD R^2 to construct the edge weights. Nonetheless, the current GNN model can be further refined with other settings (e.g., paired-wise association results from the epistatic analysis). We believe the changes will be crucial for optimizing model performance and gaining a better understanding of how polygenic risk affects human diseases. However, we hope the reviewer agrees that these analyses are beyond the scope of this study. We did include some discussion in our manuscript regarding our selection of models for the analysis (**lines 506–510**).

Table 1. Performance of graph neural network models with different edge weights

Edge weights	Number of epochs	Best accuracy in training sets	auROC in testing set
R^2	198	0.6712	0.6674
$-R^2$	219	0.6690	0.6664
$1-R^2$	178	0.6689	0.6654
$\frac{1}{2}$ -norm distance	134	0.6667	0.6658
1-norm distance	328	0.6736	0.6592

2-norm distance	242	0.6687	0.6666
Normalized 1-norm distance	179	0.6664	0.6668
Normalized 2-norm distance	164	0.6677	0.6622
Normalized $\frac{1}{2}$-norm distance	141	0.6661	0.6615
Normalized $\frac{1}{2}$-norm + (1-R²)	156	0.6674	0.6655
Normalized $\frac{1}{2}$-norm \times (1-R²)	143	0.6678	0.6625
Harmonic mean of normalized $\frac{1}{2}$-norm and (1-R²)	230	0.6682	0.6658
Normalized $\frac{1}{2}$-norm \times sqrt(1-R²)	209	0.6669	0.6611
Normalized 1-norm \times sqrt(1-R²)	134	0.6685	0.6628

auROC, area under the receiver operating characteristic curve.

Also, if possible, run and compare the neural network models using 216 GWAS SNPs or randomly select 37 SNPs to see if those 37 variants are really Chinese-specific.

We thank the reviewer for the valuable comments. In our revised manuscript, we added a new analysis by conducting the 5-fold cross-validation analysis using neural network models in the Chinese population with the following three variant groups:

- Thirty-seven variants associated with AD in the Chinese population, that were originally used in our study
- 216 GWAS AD variants that were found in the Chinese population
- Ten sets of 37 variants that were randomly selected from the 216 GWAS variants

Our data demonstrated that the 37 variants associated with AD in the Chinese population (used in the original study) exhibited a higher classification accuracy than that of the 216 variants or any set of 37 variants that were randomly selected from the 216 variants (**Figure 1**, or **Supplementary Figure 17** in the revised manuscript). We believe these results are reasonable, since only those 37 of the 216 variants are significantly associated with AD in the Chinese population. This finding also suggests that proper filtering of variants is crucial for constructing deep learning models for polygenic risk assessment.

Accordingly, we have added this result in our revised manuscript (**lines 311–314**).

Figure 1. Comparison of neural network models using different sets of variants on Alzheimer’s disease classification in the Chinese population using 5-fold cross-validation. Dot plots show the classification accuracy of neural network models constructed based on: all 216 AD variants reported by genome-wide association studies, 10 sets of 37 variants randomly selected from those 216 variants (“Randomly selected”), and the 37 AD variants that show a significant association in the Chinese population (“AD-associated”). *** $P < 0.001$ for one-way ANOVA followed by Tukey’s post hoc test comparing (a) auROC and (b) auPRC with all other variant groups. AD, Alzheimer’s disease; auPRC, area under the precision-recall curve; auROC, area under the receiver operating characteristic curve.

Here are my comments about how they addressed Rev 3.

1. Benchmarking state-of-the-art PRS methods. It is great to see they compared their weighted PRS model with several state-of-the-arts, which do not show any significant improvement over weighted PRS for classifying AD in European cohorts. However, is it possible to show the comparison on Chinese cohorts, which are the main selling point of this work?

We are grateful for the reviewer’s critical comments and suggestions.

We conducted additional analysis and did a benchmark comparison for the weighted PRS and several types of modified PRS models (including AnnoPred, LDPred, SBayesR, and Winner’s curse models) for the 216 GWAS AD risk variants identified by existing AD GWAS studies, as well as the 37 variants selected from the 216 GWAS AD risk variants due to their association with AD in the Chinese population (see **Figure 2**, or **Supplementary Figure 12** in the revised manuscript). Despite the variation in the auROC and auPRC values across different variant sets and models, their 95% confidence intervals (CIs) could not be clearly separated. Thus, we have drawn the following conclusions:

1. There are small and insignificant differences in the classification accuracy when comparing the models constructed based on 216 or 37 variants, or when comparing different wPRS models using the same set of variants.
2. SBayesR based on 37 variants achieves a slightly better classification accuracy at auROC of 0.6881 (95% CI 0.6652–0.7110). Its performance is slightly lower than that of the lasso model (auROC of 0.7069, and 95% CI 0.6845–0.7294), and is significantly lower than that of the neural network model (auROC of 0.7718, and 95% CI 0.7510–0.7925).

Our current results in the Chinese population confirm our findings in the European-descent data, i.e., compared to the weighted PRS models, the modified weighted PRS models did not significantly improve their performance in predicting AD. However, we speculate that including more variants (e.g., 10,000 variants) will increase the likelihood of the improved performance of polygenic risk analysis in AD classification. The current study is merely to investigate whether deep learning is applicable for polygenic risk analysis.

Accordingly, we have added this result in our revised manuscript (**lines 276–278**).

Figure 2. Comparison of the classification accuracy of modified polygenic risk score models using 37 variants in the Chinese population. (a–b) Performance of the different weighted polygenic risk score models in disease classification accuracy in the Chinese population using different variant sets measured by (a) auROC and (b) auPRC. (c–d) Comparison of disease classification accuracy between different models. (c) auROC and (d) auPRC values are plotted as means with error bars denoting 95% confidence intervals. auPRC, area under the precision-recall curve; auROC, area under the receiver operating characteristic curve; wPRS, weighted polygenic risk score.

2. Addressed.

3. The authors showed comparable performances between p-val select SNPs and “37 SNPs”. This might be acceptable since 37 SNPs are much less dimensional (more represented features) than thousands of SNPs. Does the latter overlap or include the former? It is unclear if the authors have also fully optimized the neural network for p-val select SNPs (e.g., Fig S14) for a fair comparison. Actually, it is very surprise to see the neural network model with much less features (N=37) is significantly better than the ones with much more features. Moreover, how did the authors generate 37 variants from European cohort (Fig. S15)? Are they the same for Chinese cohorts?

We thank the reviewers for the questions.

We selected the “thousands of SNPs” based on the AD association *p*-values obtained from an AD GWAS conducted in the European-decent population (Jansen et al., 2019; PMID 32029921). For the 37 variants, we first identified 216 AD GWAS variants reported in multiple GWASs with different ethnic groups, and found that 37 of them are significantly associated with AD in Chinese. These two sets of variants do not completely overlap (see **Table 2** below); thus, they may not capture the same genetic representation feature in disease risk. Hence, it is

interesting to speculate that the variants in the “37 variants”, that could not be found in the “thousands of SNPs”, maybe more informative in classifying AD risk in the Chinese.

Meanwhile, our previous results also showed that when comparing models with 37 versus 216 variants, the model with fewer features performs significantly better in classifying AD (**Figure 1**, or **Supplementary Figure 17** in the revised manuscript). Thus, we believe these 37 variants, selected based on Chinese AD data, are more informative in classifying AD risk in the Chinese population than the SNP sets selected by p -values from European GWAS data.

As for the “37 variants” used in the European cohort analysis (**Supplementary Figure 15**), it is the same set of 37 variants that we used for modeling the AD polygenic risk in the Chinese population (**Supplementary Table 21**).

To ensure the full optimization of the neural network models, we used a 5-fold cross-validation analysis with sufficient epochs to enable a neural network model built from 37 variants to achieve convergence in the European population data. For analysis in the Chinese data, we used the same parameter setting as we used in the European population data analysis, determining the number of epochs (1,000) by the model performance (5-fold cross-validation analysis as mentioned above) in the European population data as well as by observing the trend of model loss in the Chinese data.

In addition, we have corrected the minor errors in auROC and auPRC values when calculating the auROC and auPRC derived from the weighted PRS model using 37 variants in the Chinese population.

Table 2. The comparisons of the variants that were used for modelling Alzheimer’s polygenic risk in the European-descent population (selected by p -values) and in the Chinese population (the 37 variants)

Jansen et al. GWAS	Variants selected by p-values that overlap with the 37 variants
8,100 variants ($p < 1E-4$)	33
2,959 variants ($p < 1E-6$)	31
1,799 variants ($p < 1E-8$)	27

GWAS, genome-wide association study.

REVIEWERS' COMMENTS:

(Reviewer 4 submitted only confidential comments to the editor)